# Copper Chelation by Penicillamine Protects Against Doxorubicin-Induced Cardiomyopathy by Suppressing FDX1-Mediated Cuproptosis

**DOI:** 10.3390/biom15091320

**Published:** 2025-09-15

**Authors:** Mohammad El-Nablaway, Hany M. A. Sonpol, Yaser Hosny Ali Elewa, Mohamed A. M. Ali, Mohamed Adel, Eman Serry Zayed, Maha Alhelf, Manar A. Didamoony, Amal Fahmy Dawood, Eman M. Embaby, Khaled S. El-Bayoumi, Wesam S. El-Saeed

**Affiliations:** 1Department of Basic Medical Sciences, College of Medicine, Al-Maarefa University, Dariyah 13713, Saudi Arabia; 2Department of Medical Biochemistry and Molecular Biology, Faculty of Medicine, Mansoura University, Mansoura 35516, Egypt; wesam@mans.edu.eg; 3Department of Anatomy, College of Medicine, University of Bisha, P.O. Box 551, Bisha 61922, Saudi Arabia; hsonbol@ub.edu.sa; 4Department of Histology and Cytology, Faculty of Veterinary Medicine, Zagazig University, Zagazig 44511, Egypt; 5Faculty of Veterinary Medicine, Hokkaido University, Sapporo 060-0808, Japan; 6Department of Biology, College of Science, Imam Mohammad Ibn Saud Islamic University (IMSIU), Riyadh 11623, Saudi Arabia; mamzaid@imamu.edu.sa; 7Department of Medical Physiology, Faculty of Medicine, Mansoura University, Mansoura 35516, Egypt; madel@mans.edu.eg; 8Department of Medical Physiology, Faculty of Medicine, Mansoura National University, Gamasa 35712, Egypt; 9Department of Clinical Biochemistry, Faculty of Medicine, University of Tabuk, Tabuk 71491, Saudi Arabia; ezayed@ut.edu.sa; 10Biotechnology School, Nile University, Giza 12588, Egypt; drmahasalah@kasralainy.edu.eg; 11Medical Biochemistry and Molecular Biology Department, Faculty of Medicine, Cairo University, Cairo 11562, Egypt; 12Pharmacology and Toxicology Department, Faculty of Pharmacy, Egyptian Russian University, Cairo 11829, Egypt; manar-didamoony@eru.edu.eg; 13Department of Basic Medical Sciences, College of Medicine, Princess Nourah Bint Abdulrahman University, P.O. Box 84428, Riyadh 11671, Saudi Arabia; afdawood@pnu.edu.sa; 14Department of Physiology, Faculty of Veterinary Medicine, Mansoura University, Mansoura 35516, Egypt; emanmohamed@mans.edu.eg; 15Department of Human Anatomy and Embryology, Faculty of Medicine, Mansoura University, Mansoura 35516, Egypt; khalad200772@mans.edu.eg; 16Department of Medical Biochemistry and Molecular Biology, Faculty of Medicine, Mansoura National University, Gamasa 35712, Egypt

**Keywords:** pencillamine, cuproptosis, FDX-1, doxorubicin, cardiotoxicity

## Abstract

Background: The cardiotoxic effects of doxorubicin (DOX), a powerful chemotherapeutic drug, are widely recognized. Cuproptosis, a unique copper-dependent form of controlled cell death, may be involved in DOX-induced cardiomyopathy, according to recent findings. This study employs both in vivo and in silico procedures to investigate the protective effects of the copper chelator penicillamine (PEN) and the role of cuproptosis in DOX-related cardiotoxicity. Methods: Thirty-two adult Sprague Dawley rats were allocated into four groups (*n* = 8): control, DOX, DOX+PEN, and PEN. Cardiac function was assessed via echocardiography. Serum cardiac biomarkers (LDH, CK-MB, CTnI), oxidative stress markers (SOD, GPX, MDA), and expression levels of cuproptosis-related genes (FDX1, LIAS, SLC31A1, ATP7A) were evaluated. Histopathological examinations and immunohistochemical staining for FDX1, SLC31A1, and DLAT were performed. Molecular docking simulated PEN’s interaction with cuproptosis-related proteins. Network pharmacology and molecular docking studies were also conducted to identify core molecular targets and simulate PEN’s binding interactions with key cuproptosis regulators. Results: DOX administration induced significant cardiac dysfunction, oxidative stress, and upregulation of cuproptosis markers. PEN treatment mitigated these effects, improved cardiac function, reduced fibrosis, and suppressed the expression of cuproptosis-related *genes* and proteins. Docking results confirmed strong interactions between PEN and cuproptosis-regulatory proteins. Network pharmacology revealed 14 key overlapping targets linking PEN with cuproptosis and DOX-induced cardiotoxicity. Conclusion: This study provides experimental evidence implicating cuproptosis in DOX-induced cardiomyopathy. PEN exerts cardioprotection, potentially by targeting this pathway, offering a promising therapeutic strategy.

## 1. Introduction

Chemotherapy-induced cardiotoxicity has become a significant concern in oncology, largely due to the rising number of long-term tumor survivors [1]. The main clinical condition of this toxicity is a dose-dependent cardiomyopathy resulting in chronic heart failure, commonly observed with anthracycline-based treatment [2]. The anthracycline class drug, doxorubicin (DOX), is an effective chemotherapeutic medication that can be employed to treat various cancers, including breast cancer, sarcoma, and lymphoma [3]. DOX is an inhibitor of the DNA topoisomerase II enzyme and causes damage to the DNA [4]. However, the clinical application of DOX is extremely restricted due to its adverse drug reactions with cardiotoxicity. It has been revealed that the chief long-term adverse consequence of DOX is a higher risk of cardiotoxicity, which raises the morbidity and death rate among cancer survivors [5]. Interestingly, it has been reported that mitochondrial dysfunction, such as impaired bioenergetics, mitochondrial membrane potential depolarization, and increased ROS production, is a hallmark of DOX-induced cardiotoxicity [1,6].

Ion homeostasis in cardiac myocytes is central to myocardium integrity maintenance, contractile performance, and cardiac electrophysiology dynamics [7]. Copper is a crucial micronutrient with a twofold function. Under typical physiological conditions, it plays a functional regulatory role in remodeling the extracellular matrix, neurotransmitter metabolism, and redox equilibrium and participates in mitochondrial oxidative phosphorylation. However, copper dysregulation contributes to the pathophysiology of several diseases [8]. Collagen deposition, cardiac fibrosis, and impaired angiogenesis are hallmarks in patients with copper deficiency. The metalloallosteric nature of copper promotes binding to passive binding sites in proteins and modulates different pathways [9]. Cuproptosis is a distinct cell death mechanism that varies from apoptosis, pyroptosis, and ferroptosis. It occurs when a buildup of copper inside the body becomes toxic to cells. This copper overload specifically targets and damages key proteins within the cell’s “powerhouses,” the mitochondria. The result is a cascade of events that ultimately leads to cell death [10]. Thirteen key modulator genes were identified as cuproptosis-related genes, namely ferredoxin 1 (FDX1), solute carrier family 31 member 1 (SLC31A1), lipoyl transferase 1 (LIPT1), lipoic acid synthetase (LIAS), di-hydrolipoyl dehydrogenase (DLD), dihydrolipoamide branched chain transacylase E2 (DBT), glycine cleavage system protein H (GCSH), dihydrolipoamide S-succinyltransferase (DLST), dihydrolipoyl acetyl transferase (DLAT), pyruvate dehydrogenase A1 (PDHA1), pyruvate dehydrogenase B (PDHB), ATPase copper transporting alpha and beta (ATP7A and ATP7B) [11]. FDX1 is situated on chromosome 11q22 and encodes a low molecular weight protein that contains iron-sulfur clusters as a redox-active component. FDX1 mediates two central processes, mitochondrial cytochrome P450 reduction, which governs copper ionophore-induced cell death [12]. FDX1 can generate more toxic Cu^+^, activate cellular stress and induce protein aggregation (e.g., DLAT oligomerization) and mitochondrial impairment, eventually resulting in cuproptosis [13]. Therefore, it has been reported that FDX1 is the primary molecule that induces cuproptosis [12]. SLC31A1 primarily regulates intracellular copper influx in the small intestine’s enterocytes, where enterocytes absorb it. Excess copper is exported by the ATPase copper transporter 7A or 7B (ATP7A/ATP7B) [14,15], both of which translocate from the trans-Golgi network to the selected cellular membrane domain (basolateral or apical) to remove excess copper from the cell [16]. Disruption of this balance (e.g., SLC31A1 overexpression) leads to copper accumulation, FDX1 activation, and cuproptosis [14,17]. Lipoylation further relates copper toxicity to cell death. LIAS produces lipoylated proteins acting as copper-binding scaffolds, while FDX1 activates copper redox. Together, these processes dampen iron-sulfur cluster proteins and increase oligomerization of DLAT [18]. Moreover, it induces mitochondrial malfunction, culminating in cuproptosis [15,19].

The correlation between copper and cardiovascular diseases has shown erratic results. High serum copper was correlated with cardiovascular risk [20,21]. Cuproptosis has the potential to be a novel therapeutic mechanism for heart failure (HF), and FDX1 may be a key target for cuproptosis-based treatment of HF [22]. Consequently, we hypothesize that cuproptosis could be a potential therapeutic target for DOX-induced cardiotoxicity.

Interestingly, it has been revealed that copper chelators, such as ammonium tetrathiomolybdate and D-penicillamine (PEN), probably have an important role in inhibiting the cuproptosis pathway [23,24]. The current study selected PEN as its standard copper chelating agent, and its rapid effect on copper excretion in urine is well documented [25]. A previous investigation into the influence of PEN on catecholamine-induced myocardial damage showed the protective benefits of PEN against acute cardiac injury. Říha and his colleagues reported that this cardioprotective role may be due to PEN’s copper and/or iron homeostasis modulation [26]. PEN’s involvement in Fenton reaction dynamics mirrors the behaviors of certain reducing antioxidants, indicating a potential function in mitigating oxidative stress pathways. Through this investigation, we aspire to contribute to understanding novel preventive strategies for DOX-induced cardiomyopathy, emphasizing the role of cuproptosis. This could pave the way for developing more protective agents and potentially improve the quality of life for individuals treated with DOX.

## 2. Materials and Methods

### 2.1. Experimental Animals

A total of thirty-two adult male Sprague Dawley rats, weighing 250 ± 40 g each, were obtained from the Clinical Pharmacology Department, Faculty of Medicine, Mansoura University. The rats were housed in cages and had free access to food and water at suitable temperatures and a 12 h light–dark cycle. The rats were adapted to these conditions for at least one week before commencing the experiments, while the general conditions were monitored throughout the study. Animal experimentation complied with the National Institutes of Health’s Guide for the Care and Use of Laboratory Animals, the U.K. Animals (Scientific Procedures) Act, international ethical standards, and ARRIVE recommendations. The research methodology and animal studies were authorized by Mansoura University’s Faculty of Medicine’s Ethics Committee for Animal Experimentation guidelines (Approval number: MU-ACUC-MED.R.23.10.27).

### 2.2. Study Design

The rats were randomly divided into four experimental groups (*n* = 8): (1) The control group: received six intraperitoneal (i.p.) injections of 0.9% saline with a 48 h interval for 2 weeks; (2) The DOX group (DOX): received six i.p. injections of commercially available DOX (Hikma, Giza, Egypt) with a dose of 2.5 mg/kg body weight (bw) with a 48 h interval for 2 weeks (i.e., 15 mg/kg cumulative dose). Timing and dosing of DOX cycles were calculated based on preclinical protocols, and human chemotherapy regimens were adjusted to the lifespan, body surface area, and metabolism of rats [27]; (3) The DOX+PEN group received six i.p. injections of DOX as described in group 2 with PEN (Sigma-Aldrich, St. Louis, MO, USA) in a dose of 11 mg/kg/day, injected via tail vein; (4) The PEN group received only D-penicillamine as described in group 3. The therapeutic dose of PEN was chosen based on a previous study [26] and a pre-experimental pilot study on multiple doses (5.5 mg/kg bw, 11 mg/kg bw, 22 mg/kg bw, 44 mg/kg bw), which revealed that the medium dose (11 mg/kg bw) provided maximum therapeutic benefits on cardiac oxidative stress markers and serum cardiac marker levels, with minimal toxicity. Two days after the last treatment administration, the rats were kept under deep anesthesia (following intraperitoneal injection with ketamine and xylazine at doses of 90 mg/kg bw and 10 mg/kg bw, respectively) and examined ultrasonographically, followed by the collection of the rats’ hearts and blood from different experimental groups for further processing.

### 2.3. Rat Cardiac Echocardiography

Transthoracic echocardiography was performed using a 10 MHz phased array transducer (Aplio ver. 6.0; Toshiba, Tokyo, Japan). M-mode echocardiography was used to visualize the papillary muscles in a parasternal short-axis perspective. Fractional shortening (FS) was computed using the formula FS = [(left ventricle “LV” internal diameter at end-diastole) − (LV internal diameter at end-systole)/LV internal diameter at end-diastole] × 100. Teichholz’s formula [28] was used to compute the ejection fraction using the end-systolic and end-diastolic volumes.

### 2.4. Determination of Heart Weight (HW) to BW Changes in DOX-Induced Cardiomyopathy (DiCM) Model

Rats were sacrificed by cervical dislocation, and the BW was recorded. Then, the chest was opened, and the heart was immediately removed and washed thoroughly with ice-cold 0.9% sodium chloride solution (saline) and dried with filter paper before weighing to measure HW. The HW/BW ratios were calculated. After removal of the atria and large vessels of the base, the left ventricle was transversely sectioned into 3 parts of 3 mm thickness between apex and base; the first part of cardiac tissue was immediately fixed in 4% paraformaldehyde (PFA) for further histopathological examination (including Hematoxylin/Eosin (HE) and Masson’s trichrome (MT) staining and immunohistochemical staining). The other parts were used to assess biochemical parameters. The second portion of cardiac tissue measured antioxidant markers and lipid peroxidation in tissue homogenates. The third portion of cardiac tissue quantitatively measured cuproptosis-dynamic genes.

### 2.5. Determination of Cardiac Biomarkers in the DiCM Model

The blood samples were collected from the heart of each rat and kept in EDTA-free tubes. Blood was allowed to coagulate at room temperature before being centrifuged at 3000 rpm for 15 min (Hettich universal 32A, Kirchlengern, Germany) to extract serum. Finally, serum samples were kept in aliquots at −20 °C until the analysis was completed. The serum levels of lactic dehydrogenase (LDH), creatine kinase MB (CK-MB), and cardiac troponin I (CTnI) were measured using commercially available kits: LDH assay endpoint kit (MG; Egypt, cat. no. MG283001), CK-MB ELISA kit (Elabscience, Houston, TX, USA, CAT: E-CL-R0722), and CTnI ELISA kit (Elabscience, Houston, TX, USA, CAT: E-EL-R1253). The absorbance was measured at 450 nm by a Sunrise absorbance microplate reader (TECAN, Männedorf, Switzerland). The concentrations were calculated according to the manufacturer’s specifications [29].

### 2.6. Preparation of Tissue Samples and Homogenate

The cardiac tissues were cleaned with cold-buffered saline 0.9%. They were used for further biochemical analysis of oxidative stress, ELISA, and PCR. Briefly, the tissues were homogenized in 1.15% KCl (pH 7.4) ice-cold medium employing a hand-held homogenizer (Omni International, Kennesaw, GA, USA). The supernatants were used for additional analysis after being collected by centrifugation at 1000× *g* for 15 min [30].

### 2.7. Analysis of Oxidative Stress in Tissue Homogenate

Tissue homogenates were used to measure the activity levels of antioxidant markers, superoxide dismutase (SOD) and glutathione peroxidase (GPX), and lipid peroxidation indicator malondialdehyde (MDA) using the ELISA kits (MyBioSource, San Diego, CA, USA, MBS036924, MBS744364, MBS268427, respectively). The absorbance was measured at 450 nm by a Sunrise absorbance microplate reader (TECAN, Männedorf, Switzerland). They were determined using the previously described methods [31,32].

### 2.8. Quantitative Real-Time PCR (qRT-PCR) for Cuproptosis- Dynamic Genes (FDX1, LIAS, SLC31A1, and ATP7A) mRNA Assay

Cardiac tissue samples were preserved in RNA later RNA Stabilization Reagent, approximately 10 μL of the reagent per 1 mg of tissue sample (Qiagen, Hilden, Germany), kept overnight at 2–8 °C, then stored at −80 °C until homogenization of tissue samples by five strokes of liquid nitrogen. The RNA was isolated using the QIAzol Lysis Reagent kit (Qiagen, Hilden, Germany). The RNA concentration and purity of the samples were assessed using the NanoDrop 2000c Spectrophotometer (Thermo Scientific, Waltham, MA, USA). The reverse transcription of 1 µg of RNA was performed using the SensiFAST cDNA Synthesis Kit (Bioline, London, UK). The RT-PCR assays were performed on the 7500 Real Time PCR System (Applied Biosystem, Waltham, MA, USA). The PCR reaction contained a total reaction volume of 20 µL (10 µL of HERAPLUS SYBR^®^ Green qPCR Master Mix, 2 µL of the synthesized cDNA, 1 µL of forward primer, 1 µL of reverse primer, and the remaining 6 µL was RNase-free water) [33]. Primer sets were synthesized by Vivantis (Vivantis Technologies, Shah Alam, Malaysia). The primer sequences used for qPCR are listed in Table 1. Glyceraldehyde-3-phosphate dehydrogenase (GAPDH) was applied as a housekeeping gene. The primer sets were designated using Primer 3 software (v.4.1.0), and primer specificity was determined using the Primer-BLAST program. The PCR reactions were conducted with the following program:2 min at 95 °C, followed by forty cycles of denaturation at 95 °C for 10 s and annealing/extension at 60 °C for 30 s. The reaction’s melting curve was analyzed to evaluate the products’ specificity. Relative quantification (RQ) of mRNA expression was estimated using the equation RQ = 2^−ΔΔCt^ [34].

### 2.9. Network Pharmacology-Based Target Identification Screnning

A network pharmacology approach was employed to elucidate the potential molecular mechanisms by which Penicillamine (PEN) may mitigate doxorubicin (DOX)-induced cardiotoxicity. Potential protein targets of PEN were predicted using three robust in silico platforms: SwissTargetPrediction, Similarity Ensemble Approach (SEA), and Polypharmacology Browser 2 (PPB2). These tools leverage chemical structure similarity, pharmacophore modeling, and ligand-based prediction algorithms to identify biologically relevant targets.

In parallel, genes associated with DOX-induced cardiotoxicity and cuproptosis were retrieved from multiple comprehensive biomedical databases, including GeneCards, DisGeNET, Comparative Toxicogenomics Database (CTD), PharmGKB, and the Therapeutic Target Database (TTD). Search queries included terms such as “cuproptosis,” “cardiotoxicity”, and “doxorubicin-induced cardiomyopathy.” All gene sets were filtered for Homo sapiens and standardized using UniProt identifiers to ensure consistency across datasets.

Overlapping targets between PEN-predicted proteins and disease-related genes were identified using Venn diagram analysis, representing potential nodes through which PEN may exert therapeutic effects. These common targets were integrated into a drug–target–disease network, constructed and visualized in Cytoscape (v3.9.1), enabling the identification of key nodes and interrelations indicative of multi-target effects.

To evaluate functional associations among overlapping targets, a Protein–Protein Interaction (PPI) network was generated using the STRING database, filtered for Homo sapiens, and a minimum interaction score of 0.4. The resulting network was imported into Cytoscape for visualization and analyzed using the CytoHubba plugin, which identified central hub genes based on topological features. Further validation and visualization of gene relationships were performed using GeneMANIA (https://genemania.org/, accessed on 5 August 2025), integrating various biological data types, including physical and genetic interactions, co-expression patterns, shared pathways, and protein domain similarities. Gene function prediction and pathway relevance were also assessed through the GeneMANIA (https://genemania.org/, accessed on 5 August 2025) platform.

Gene Ontology (GO) enrichment was performed for Biological Processes (BP) to characterize the biological significance of the intersecting genes. In addition, KEGG pathway analysis was conducted using Shinygo 0.82, providing insight into the associated.

### 2.10. Molecular Docking Screnning

To investigate the molecular interactions between Penicillamine (PEN) and selected cuproptosis-related targets implicated in DiCM, a structure-based molecular docking approach was employed using AutoDock4.2 and AutoDock Tools (ADT, version 1.5.6) [35].

The three protein targets selected for docking were ferredoxin 1 (FDX1, PDB ID: 3P1M), dihydrolipoamide acetyltransferase (DLAT, PDB ID: 3B8K), and Solute carrier family 31 member 1 (SLC31A1, PDB ID: 2LS2), all retrieved from the RCSB Protein Data Bank (www.rcsb.org, accessed on 1 June 2025). Preprocessing of receptor proteins involved removing water molecules, adding polar hydrogens, and Kollman united atom charges using ADT [36]. Gasteiger charges were assigned, and non-polar hydrogens were merged to prepare the receptors for docking.

For validation purposes, positive control ligands were docked into their corresponding target proteins. For FDX1, FMN (flavin mononucleotide) was selected as the control ligand, given that FDX1 is a mitochondrial ferredoxin that facilitates electron transfer to cytochrome P450s and naturally interacts with steroidogenic P450 systems via FMN as a redox cofactor [37]. For DLAT, lipoic acid was chosen as the control, since the DLAT active site physiologically interacts with CoA during acetyl transfer, and lipoic acid derivatives are well-established biological ligands [38]. For SLC31A1 we employed a thiosemicarbazone compound (NSC73306), a previously reported CTR1-transported small molecule, as a biologically relevant, non-platinum control. [39].

The chemical structure of penicillamine and the 3 positive controls (FMN, lipoic acid, and thiosemicarbazone) were obtained from the PubChem database, then energy-minimized and converted to PDBQT format after adding Gasteiger charges and rotatable bonds using ADT.

Docking grid boxes were generated to encompass all the crystal structures of the three target proteins, enabling blind docking to explore all potential binding sites without prior assumptions about the active site location. The grid spacing was set to 0.375 Å, and grid maps were generated using AutoGrid4.

Docking was performed using AutoDock4’s Lamarckian Genetic Algorithm (LGA) with the following parameters: a population size of 150, 2,500,000 energy evaluations, 27,000 generations, and 100 docking runs for each target-ligand pair [40]. This approach enabled efficient sampling of ligand conformations and binding orientations.

The resulting docking poses were ranked based on binding free energy (ΔG binding in kcal/mol), and root-mean-square deviation (RMSD) was used to assess pose convergence and clustering consistency. The best-scoring pose for each protein was selected based on its lowest binding energy and favorable interactions with key residues, as evaluated visually in BIOVIA Discovery Studio Visualizer 2021 for generating 2D and 3D interaction figures [41].

### 2.11. Histopathological Examination of Cardiac Tissue by Hematoxylin/Eosin, Masson’s Trichrome, and Immunohistochemical Staining

The PFA-fixed samples were washed, dehydrated in ascending graded ethanol, cleared in xylene, and embedded in paraffin. 3-µm-thickness paraffin sections were prepared and used for HE staining (for examination of the morphological changes); MT staining (for analysis of collagen and extracellular matrix “ECM” deposition); and immunohistochemical staining for the cuproptosis dynamic protein markers (SLC31A1, FDX1, and DLAT), using anti-SLC31A1/CTR1 antibody (Cat. No. GTX48534, GeneTex, Irvine, CA, USA); Anti FDX1 Polyclonal antibody (Cat. No. 12592-1-AP, Proteintech, Chicago, IL, USA); and anti-DLAT antibody (Cat. No. CSB-PA445587, CUSABIO, Houston, TX, USA), respectively (Table 2). Following deparaffinization and rehydration, the sections were subjected to heat-induced antigen retrieval using Tris-HCl buffer saline (pH 9) at 110 °C for 15 min. Next, the sections were immersed in a solution of 0.3% H_2_O_2_/methanol for 20 min at room temperature to block the endogenous peroxidase activity. Then the sections were washed with PBS thrice for five minutes each and were incubated for 1 h at room temperature with 10% normal goat serum blocking serum. Next, the sections were incubated overnight with the specific primary antibodies at 4 °C. Then, the sections were washed thrice with PBS for five minutes each and incubated at room temperature with the anti-rabbit secondary antibody for 30 min. Subsequently, the sections were flushed with PBS and incubated for 30 min with streptavidin (SABPO (R) kit; Nichirei, Tokyo, Japan) at room temperature. Then, the sections were washed thrice with PBS/ 5 min each and subjected to short incubation with 3,3′ diaminobenzidine tetrahydrochloride-H_2_O_2_ solution until the immunopositive reactions were developed. Finally, the sections were washed in deionized water, lightly stained with Mayer’s hematoxylin, dehydrated, mounted, and examined under a BZ-X710 microscope (Keyence, Osaka, Japan)

### 2.12. Morphometrical Measurements

The percentages of collagen fiber deposition area fraction and the SLC31A1, FDX1, and DLAT immunopositive reactivity were measured using ImageJ (ver. 1.32j, http://rsb.info.nih.gov/ij, accessed on 5 August 2025) and compared among the studied groups. Three images were randomly captured from either MT or immune-stained sections of all rats’ hearts of the different studied groups and were used for further measurement by ImageJ software. Briefly, the captured images were uploaded to the ImageJ software, and the scale bar was set, and both area and area fraction were selected using the set measurement tool. Then the RGB images were processed using the color deconvolution tool. We selected Masson Trichrome and H DAB from the vector tool for the images captured from the MT and the immunostained sections, respectively. The images from the MT-stained sections were split into three constituent channels: blue, red, and green. The green channel was quantified using the “threshold” tool, following manual threshold adjustment until all positive green areas were highlighted in red. Then, the adjusted red threshold area was measured using the “measurement” button from the “analyze” tool, and the results were recorded. The average measurement of all stained sections/groups was compared among all studied groups. Similarly, for quantifying the amount of positive DAB staining in the immunostained section, the captured images were split into three channels by color deconvolution, and the brown channel was used to measure the percentages of area fraction similar to that mentioned for the measurement of the positive green area in the MT-stained section according to previous described method [42].

### 2.13. Statistical Analysis

Data analysis was carried out by GraphPad Prism 9 software. Shapiro–Wilk’s test was used to test the normality of quantitative data. The four groups’ quantitative data were compared using the one-way ANOVA test.

## 3. Results

### 3.1. PEN Attenuated Cardiac Atrophy and Improved Echocardiographic Parameters

To investigate the roles of PEN in cardiac function, we employed echocardiographic parameters and evaluated HW and HW/BW to assess how DOX and PEN administration affected cardiac performance. Intraperitoneal DOX administration produced significant cardiac atrophy, as evidenced by a notable reduction in absolute HW (676.66 ± 44.16 mg) vs. control (936.83 ± 28.81 mg) *p* value < 0.001, and a marked reduction in HW/BW ratio (2.2 ± 0.4) compared to the control group (4.06 ± 0.32 mg) *p* value < 0.001, which confirms DOX-mediated pathological remodeling, consistent with cardiomyopathy, as shown in Figure 1. In contrast, the PEN administration (908.33 ± 32.35 mg) exhibits a non-significant impact relative to the control, and PEN-DOX co-administration (815 ± 47.64 mg) significantly alleviates the effects of DOX toxicity, restoring absolute HW ratios to levels nearly comparable to baseline (*p* < 0.001). The HW/BW ratio in the PEN group (3.98 ± 0.40) and the PEN-DOX co-administration group (3.18 ± 0.38) approximated similar to the initial values, delineating the reversal impact of DOX-induced cardiotoxicity, suggesting PEN has potential in amelioration of structural myocardial damage by alleviating the HW/BW ratio. Echocardiographic analysis of left ventricular function demonstrated severe impairment in the DOX-induced cardiotoxicity, detailed by dramatic reductions in fractional shortening (FS) (35.96 ± 4.76) compared to control (52.05 ± 3.04), *p* value < 0.001. In contrast, the ejection fraction of the left ventricle (LVEF) of the DOX-administered group showed 62.35 ± 5.18 relative to controls 87.88 ± 0.43 (*p* < 0.001). These findings confirmed the successful induction of chronic DOX-mediated cardiotoxicity. Interestingly, PEN exhibits a non-significant impact on FS (55.11 ± 4.51) relative to the control (52.05 ± 3.04). Most importantly, the DOX-PEN co-administered group showed substantially ameliorated FS, exhibiting significantly higher FS values (45.96 ± 5.10) compared to the DOX group (*p* < 0.01). These cumulative changes in hemodynamic variables demonstrate PEN’s ability to mitigate DOX-induced myocardial damage, emphasizing its potential as a therapeutic agent for preserving cardiac structure and function during chemotherapy-induced toxicity.

### 3.2. PEN Alleviated Cardiac Functions in the DiCM Model

To explore the cardioprotective effects of PEN therapy, we evaluated serum cardiac biomarkers (LDH, CK-MB, and cTnI) using ELISA in experimental models of DOX-induced cardiotoxicity routinely used for assessing myocardial injury and provided precise information on the protective effects of PEN against DOX-induced cardiotoxicity. Our findings revealed that the DiCM displayed a considerable augmentation in the CK-MB level (1101.29 ± 31.75 U/L) compared to the control (299.86 ± 8.77 U/L). These data demonstrate that DOX has effectively confirmed pathological remodeling, consistent with cardiotoxicity. PEN administration substantially restored CK-MB levels (305.45 ± 11.55 U/L). The co-administration of DOX and PEN demonstrated a significant normalization of serum cardiac CK-MB levels (375.86 ± 47.53 U/L), compared to the DOX-administered group (*p* < 0.0001). Serum LDH revealed a substantial augmentation in the DOX administration group (1292.1 ± 23.71 U/L) relative to the control group (344.87 ± 9.38 U/L). Markedly, PEN administration (366.23 ± 18.43 U/L) exhibits a non-significant impact relative to the control (344.87 ± 9.38 U/L), indicating that PEN alone possesses a non-substantial effect on normal cardiac function. The co-administration of PEN and DOX showed a significant restoration of serum LDH level. The assessment of CTnI levels indicated that DOX administration significantly increased CTnI levels (0.071 ± 0.022 ng/mL) compared to the control group (0.019 ± 0.007 ng/mL), confirming that DOX therapy induces pathological remodeling of cardiomyopathy.

In contrast, PEN administration exhibited no significant effect compared to the control (0.018 ± 0.006 ng/mL), validating its cardioprotective properties on cardiac function. Furthermore, the co-administration of PEN and DOX resulted in a substantial restoration of CTnI levels (0.033± 0.01 ng/mL) approaching baseline values, preventing the registered effect of DOX. These findings run in parallel with the results of echocardiography parameters, proving the occurrence of cardiac damage (Figure 2). Concurrent PEN administration with DOX showed a noteworthy normalization of serum cardiac markers. It mitigated these adverse effects of DOX administration, which unequivocally supports PEN administration as a therapeutic agent for preserving cardiotoxicity.

### 3.3. PEN Treatment Alleviates DOX-Induced Oxidative Stress in DOX-Induced Cardiomyopathy

Motivated by the manipulation of the cardiac biomarkers (CK-MB, LDH, and CTnI), we extended our investigations to examine the level of SOD, GPx, and MDA enzymes as representative markers for enzymatic antioxidant activity to comprehensively estimate the oxidative status and elucidate the mechanistic pathway of PEN administration to modulate oxidative stress. Our results showed a considerable reduction in SOD levels (7.73 ± 0.61 IU/mg protein) in the DOX-administered group relative to the control (14.28 ± 0.84 IU/mg protein), suggesting oxidative stress. PEN administration (15.25 ± 0.67 IU/mg protein) showed a non-substantial effect compared to the control. Remarkably, co-administration of PEN and DOX (10.72 ± 0.51 IU/mg protein) substantially reversed oxidative stress, demonstrating a pronounced impact on the enzyme antioxidant capacity. Additionally, the DOX-administered group (100.33 ± 3.67 nM) exhibited a significant increase in MDA levels compared to the control group (53.26 ± 4.71 nM), with no notable change observed in the PEN (48.05 ± 6.59 nM). In contrast, the co-administration of PEN and DOX restored MDA levels (61.48 ± 3.97 nM) to near-normal levels. Intriguingly, the DOX-induced cardiotoxicity group demonstrated a substantial decrease in GPx level (58.85 ± 3.81 nmol/min/100 mg protein) relative to control (149.38 ± 6.78 nmol/min/100 mg protein), indicating that DOX has effectively triggered oxidative stress pathways in the cardiac tissues. While PEN (147.13 ± 9.71 nmol/min/100 mg protein) demonstrated no observed effect correlated with control, co-administration of DOX and PEN therapy showed a pronounced elevation in GPx level close to baseline levels (121.95 ± 9.92 nmol/min/100 mg protein) (Figure 3). These findings indicate that PEN exhibits a substantial impact to induce the antioxidant pathway in DOX-induced cardiotoxicity by targeting SOD, MDA, and GPx, demonstrating a more pronounced impact on the enzyme antioxidant capacity.

### 3.4. PEN Administration Mitigated the DOX-Induced Cuproptosis

To investigate whether modulation of the cuproptosis signaling axis by PEN holds therapeutic promise and to comprehend the precise mechanism by which PEN suppresses cuproptosis. We propose estimating its modulatory role in the DOX-administered model to activate cuproptosis and understand the distinct molecular mechanisms underlying PEN- and DOX-induced cardiotoxicity, focusing on copper-dependent cell death pathways. The analysis confirmed a trend toward elevated expression levels of cuproptosis-related genes (FDX1, LIAS, SLC31A1) in rats subjected to DOX, accompanied by reduced ATP7A expression levels, which was mitigated by PEN administration, as shown in Figure 4. Intraperitoneal DOX administration exacerbates intracellular copper overload by forming redox-active copper complexes, which amplify oxidative stress and mitochondrial damage. Upregulated key cuproptosis-related genes FDX1 (3.89 ± 0.77), LIAS (3.2 ± 0.8), and SLC31A1 (4.72 ± 1.05) and dampened ATP7A (0.335 ± 0.06) expression compared to control, consistent with its role in disrupting copper metabolism, which indicates that DOX administration effectively promotes pathological remodeling of cardiotoxicity.

On the other hand, the PEN group showed no significant effect on cardiac cuproptosis genes FDX1 (1.66 ± 0.43), LIAS (1.45 ± 0.21), SLC31A1 (1.56 ± 0.39), and ATP7A (1.16 ± 0.35). At the same time, co-administration of PEN and DOX likely interrupts this cascade by binding free copper ions, thereby preventing DOX-copper complex formation and subsequent cuproptosis, and normalization of FDX1 (1.63 ± 0.53), LIAS (2.05 ± 0.61), SLC31A1 (2.31 ± 0.83), and ATP7A (1.25 ± 0.42) expression nears baseline levels, where it reduces systemic copper toxicity (Figure 4). This mechanism proposed copper chelation, as PEN protects against DOX cardiotoxicity by stabilizing copper pools. PEN’s modulation of the cuproptosis signaling axis further reflects the cell’s attempt to restore copper balance under DOX stress.

### 3.5. Histological Studies

We examined the HE-stained cardiac sections among the different studied groups to investigate whether PEN injection improves cardiac morphology in DOX-induced cardiomyopathy (Figure 5). The HE-stained cardiac sections in both control and PEN groups revealed normal myocardial structure. The blood vessels were lined with flat endothelial cells and a thin basal lamina (Figure 5A,D). In contrast, the DOX group showed pronounced myocardial atrophy with the disruption of actin-myosin striation and fewer intercalated disks. In addition, our examination revealed a thick basal lamina surrounding the lining endothelium (Figure 5B). Notably, in the DOX+PEN group, there was a notable improvement in the aforementioned pathological structure, particularly the actin-myosin striation and thickness of the basal lamina (Figure 5B).

Furthermore, examination of the MT-stained cardiac sections revealed a notable increase in the interstitial collagen fibers in the DOX group (Figure 6B) compared with the control and PEN groups (Figure 6A,D). Interestingly, the co-administration of PEN with DOX apparently showed a reduction in the interstitial collagen fiber deposition (Figure 6C) compared to the DOX group (Figure 6B). We measured the percentage of positive collagen fiber area fraction among the studied groups to detect the degree of fibrosis. Intriguingly, a significantly higher degree of fibrosis was observed in the DOX group compared to the control and PEN groups. Furthermore, the DOX+PEN group showed a significant reduction in fibrosis degree compared with the DOX group, with no significant difference between the control and PEN groups (Figure 6E).

### 3.6. Immunohistochemical Analysis

To examine the impact of PEN on DOX-provoked changes in the expression level for cuproptosis-related genes, an immunohistochemical study was performed using Slc31A1 (Figure 7), FDX1 (Figure 8), and DLAT (Figure 9) antibodies. The cardiac muscle in the control and PEN groups revealed faint positive cytoplasmic reactions for the SLC31A (Figure 7A,D) and FDX1 (Figure 8A,D) and a few positive nuclear reactions for the DLAT immunostained sections (Figure 9A,D). On the other hand, the cardiac sections of the DOX group revealed a strong positive cytoplasmic reaction in the SLC31A (Figure 7B) and FDX1 immuno-stained sections (Figure 8B) and numerous positive nuclear reactions for the DLAT immuno-stained sections (Figure 9B). Notably, PEN treatment following DOX administration induced a reduced accumulation of cytoplasmic SLC31A (Figure 7C) and FDX1 (Figure 8C) and less nuclear expression of DLAT (Figure 9C) in the PEN+DOX group. Similarly, our quantitative data for the percentage of positive area fractions of the expression level for the studied cuproptosis-related genes revealed highly significant values in the DOX group compared to other studied groups. Moreover, as compared to both control and PEN groups, the PEN+DOX group revealed non-significant, significant, and highly significant differences for the Slc31A1, FDX1, and DLAT, respectively (Figure 7E, Figure 8E, and Figure 9E).

### 3.7. Network Pharmacology-Based Target Identification

Target prediction for PEN using SEA, SwissTargetPrediction, and PPB2 identified multiple candidate proteins. In parallel, 3740 non-redundant genes associated with DOX-induced cardiotoxicity and cuproptosis were collected from the integrated databases.

Comparative analysis showed that PEN shared approximately 0.9% of targets with cardiotoxicity/cuproptosis genes via SEA prediction, 0.6% via PPB2, and 0.9% via SwissTargetPrediction, with a total overlap of 1.7% across all methods. Intersection analysis revealed 14 (0.4%) shared genes between the predicted PEN targets and disease-associated genes (Figure 10); namely, ACE, ATP7A, DLAT, FDX1, NOS1, NOS2, SLC31A1, CA2, ODC1, CA9, ANPEP, SLC1A2, EGLN1, and CA1. These overlapping targets represent key mediators potentially involved in PEN’s cardioprotective mechanisms via cuproptosis modulation.

The shared targets generated a PPI network via STRING, revealing 44 nodes and 222 edges (Figure 11A). Topological analysis in Cytoscape using CytoHubba identified key hub genes central to the network structure (Figure 11B).

GeneMANIA analysis further elucidated gene–gene interactions and predicted functions based on diverse biological datasets (Figure 11C). This network incorporated data from physical interactions, co-expression, pathway associations, and protein domains, supporting the involvement of the identified genes in relevant cardiotoxicity and cuproptosis-related pathways.

### 3.8. KEGG and GO Pathway Enrichment

The KEGG pathway enrichment analysis revealed several biologically meaningful pathways that explain the cardioprotective effects of penicillamine against doxorubicin-induced toxicity. The top significant pathways are shown in a barplot in Figure 12A.

Among the top enriched pathways, the renin-angiotensin system (RAS) (FDR = 8.0 × 10^−4^) emerged as particularly relevant (Figure 12B). The RAS is pivotal in cardiovascular regulation, influencing vasoconstriction, fluid retention, and cardiac remodeling. Disruption of this pathway has been implicated in DOX-induced cardiac injury, where elevated angiotensin II levels promote oxidative stress, apoptosis, and fibrosis. Penicillamine may modulate this system by influencing components such as angiotensin-converting enzyme (ACE), thereby mitigating DOX-induced cardiovascular damage.

Additionally, enrichment of the HIF-1 signaling pathway with FDR = 8.2 × 10^−3^ underscores a potential adaptive mechanism to oxidative and hypoxic stress. Hypoxia-inducible factor 1 (HIF-1) is a transcription factor that regulates genes involved in angiogenesis, glucose metabolism, and survival under low oxygen conditions. Its activation has been reported in DiCM as a compensatory response. Penicillamine may enhance or stabilize HIF-1 signaling, promoting cell survival and neovascularization to counteract tissue damage (Figure 12C).

Identifying the relaxin signaling pathway further supports the cardioprotective role of penicillamine. Relaxin, a peptide hormone, exerts vasodilatory, anti-inflammatory, and anti-fibrotic effects in the cardiovascular system. This pathway’s enrichment suggests that penicillamine could mimic or potentiate relaxin’s effects, thus alleviating myocardial stiffness and improving vascular compliance in DOX-treated models (Figure 12D). Likewise, the apelin signaling pathway, known for its role in cardiac contractility, fluid homeostasis, and inhibition of apoptosis, was among the enriched categories. Activation of this pathway is associated with cardioprotective effects, particularly in the context of ischemic injury and heart failure. The modulation of Apelin signaling by Penicillamine could improve cardiac output and reduce cell death in the presence of DOX (Figure 12E).

Gene ontology biological processes (GO: BP) enrichment analysis offers valuable insights into the potential biological mechanisms by which penicillamine may counteract DiCM, like oxidative stress response, apoptosis regulation, mitochondrial dysfunction, and copper ion homeostasis (Figure 13). Among the most significantly enriched terms were protein nitrosylation and peptidyl-cysteine S-nitrosylation, both pointing toward modulating redox-sensitive signaling pathways. DOX-induced oxidative stress leads to aberrant nitrosative modifications of proteins, disrupting their function and contributing to cardiomyocyte apoptosis. Penicillamine, a thiol-containing chelator, may act by buffering reactive nitrogen species and preserving thiol groups in protein cysteine residues, thus attenuating these damaging modifications.

Several copper-related processes—including copper ion import, transmembrane transport, and cellular copper ion homeostasis—were also highly enriched. Copper is an essential cofactor for numerous antioxidant enzymes, including SOD, and its dysregulation is linked to oxidative tissue damage. Penicillamine is known to chelate copper, suggesting that its cardioprotective role may also involve normalization of copper homeostasis under DOX-induced oxidative stress conditions.

Another key process identified was positive regulation of guanylate cyclase activity, which plays a vital role in cardiovascular homeostasis by mediating nitric oxide (NO) signaling. This pathway promotes vasodilation and protects against ischemia–reperfusion injury, which are common features of DiCM. Penicillamine could support myocardial perfusion and reduce hypertensive stress by facilitating this process.

Enrichment in arginine metabolic and catabolic processes also aligns with the role of nitric oxide synthase (NOS) enzymes, which use arginine to produce NO. DOX impairs NOS function, leading to a reduced NO bioavailability. Penicillamine may help maintain or restore arginine-NO signaling, enhancing endothelial function and limiting vascular damage.

Enriching responses to hypoxia and decreased oxygen levels further support a potential adaptive mechanism mediated by penicillamine. These responses typically involve HIF-1α activation and metabolic reprogramming that favor cell survival under oxidative and hypoxic conditions. As DOX compromises mitochondrial respiration and oxygen delivery, pathways promoting adaptation to hypoxia are critical for preserving cardiac tissue integrity.

Moreover, terms such as “blood pressure regulation,” “circulatory system processes,” and “trans-synaptic/synaptic signaling” emphasize the systemic physiological responses potentially influenced by PEN. These may include improved vasoregulation, neurotransmitter balance, and intercellular communication—each contributing to cardiovascular stability under stress.

Enriching broad homeostatic processes—including cellular, ion, chemical, and metal ion homeostasis—highlights PEN’s multifaceted role in maintaining physiological equilibrium. DOX disrupts ionic gradients, redox states, and metabolic fluxes; hence, penicillamine’s ability to stabilize these parameters likely underlies its protective efficacy.

### 3.9. Molecular Docking

To complement the network pharmacology findings and further elucidate the potential cardioprotective mechanism of Penicillamine (PEN) against DOX-induced cardiotoxicity, molecular docking simulations were conducted targeting three key cuproptosis-associated proteins identified in the pharmacological interaction network: ferredoxin 1 (FDX1, PDB ID: 3P1M), dihydrolipoamide acetyltransferase (DLAT, PDB ID: 3B8K), and solute carrier family 31 member 1 (SLC31A1, PDB ID: 2LS2).

The top-scoring pose for each target was selected based on binding affinity and quality of interaction with active site residues. The results are summarized in Table 3 and illustrated in Figure 14.

PEN demonstrated favorable binding affinity to all three targets, with docking scores ranging from –3.53 to –4.45 kcal/mol (Table 3). Among the targets, DLAT exhibited the most favorable binding affinity, with a docking score of −4.45 kcal/mol, followed by FDX1 (−3.89 kcal/mol) and SLC31A1 (−3.53 kcal/mol). These values indicate a moderate yet biologically plausible binding of PEN to the respective targets, supporting its potential to modulate cuproptosis-related pathways.

For FDX1, PEN formed multiple stabilizing interactions with Asn73 (hydrogen bond, 2.11 Å), Arg74 (salt-bridge, 2.33 and 1.92 Å and H-bond, 2.67 Å), and Asp75 and Asp101 (ionic bonds at 3.02 Å and 2.22 Å, respectively). These interactions are localized near regions implicated in the electron transfer function of FDX1, suggesting that PEN may disrupt or modulate its redox activity (Figure 14B). As shown in Figure 14A, the positive control (FMN) engaged the same key residues with identical interaction patterns, except for an additional hydrogen bond with Asp173, confirming that PEN binds in a functionally relevant manner similar to the control.

In the case of SLC31A1, hydrogen bonding was observed with Met19 at distances of 3.02 Å and 2.01 Å, as well as with Leu24 (3.05 Å), indicating possible stabilization at the copper transport interface of this membrane protein. These findings imply that PEN may interfere with copper ion uptake, consistent with its role as a copper chelator (Figure 14D). Furthermore, thiosemicarbazone (positive control) was able to bind within the same pocket (Figure 14C), interacting with Met19 and Leu24 through hydrophobic contacts while forming a hydrogen bond with Gly23. This overlap in binding interactions further validates the predicted binding mode of PEN.

For DLAT, the strongest binding was observed through a salt-bridge interaction with Arg535 (3.20 Å) and an ionic bond with Glu407 (2.30 Å). These residues are proximal to the lipoyl domain and acetyltransferase catalytic core, potentially implicating PEN in disrupting TCA-cycle-linked enzymatic activity central to cuproptosis (Figure 14F). Similarly, Lipoic acid (Figure 14E) was able to form the same salt bridge with Asp535, along with an H-bond with Asp533 and a π-sulfur interaction with Trp545, further confirming the validity of the binding site.

Collectively, these docking results highlight direct molecular interactions between PEN and key residues involved in copper metabolism and mitochondrial function. These findings align with the in vivo findings and support the proposed mechanism by which PEN mitigates DOX-induced cardiotoxicity through the inhibition of cuproptosis.

## 4. Discussion

Penicillamine (PEN), a well-known copper-chelating agent primarily used for the management of Wilson’s disease [43], has demonstrated therapeutic benefits in other conditions such as rheumatoid arthritis, scleroderma, and heavy metal poisoning [44]. Penicillamine is a cornerstone of treatment for Wilson’s disease, a hereditary disorder defined by a toxic copper overload that gives rise to a wide array of severe symptoms. The drug’s therapeutic efficacy is directly linked to its potent copper-chelating properties, which facilitate the removal of this excess copper from the body. The clinical presentation of this copper toxicity is diverse, encompassing significant hepatic damage, such as cirrhosis and liver failure, and severe neuropsychiatric symptoms, including tremors, dystonia, and depression [45]. As DOX exposure promotes mitochondrial dysfunction and oxidative stress, we postulated that elevated copper levels contribute to cardiac injury through a copper-dependent cell death mechanism known as cuproptosis. PEN, by enhancing urinary copper excretion, could mitigate these toxic effects [46,47].

Our in vivo findings confirmed that DOX administration induces substantial cardiotoxicity, characterized by reductions in heart weight/body weight ratio, left ventricular ejection fraction (LVEF), and fractional shortening (FS), alongside elevated levels of cardiac biomarkers (LDH, CK-MB and CTnI). These results align with prior studies documenting the cardiotoxic impact of anthracyclines [48,49]. Our histological examination confirmed the cardiotoxic effect of DOX on cardiac muscle, represented by pronounced myocardial atrophy and increased fibrosis level. This was consistent with previous reports [50,51]. Notably, PEN treatment significantly reversed these deleterious changes, restoring functional cardiac indices, suggesting a protective effect against DOX-induced cardiac injury. These findings coincide with those of [26], who investigated D-Pencillamine’s cardioprotective effect against catecholamine-induced myocardial injury. Interestingly, our histopathological results support the protective impact of PEN on cardiac structure and decreased degrees of fibrosis.

To delve into the mechanistic insight involved in the cardiac alterations induced by DOX administration, we evaluated the level of oxidative stress in the heart muscles. An altered membrane function due to DOX-induced lipid peroxidation might be responsible for most of the ECG changes, as it is well-known that DOX induces cardiomyopathy [52,53]. Our results corroborate these mechanisms, signifying that DOX administration substantially reduced cardiac SOD and GPx levels concomitant with elevating MDA, suggesting prominent lipid peroxidation. These observations align with prior studies reporting depletion of SOD and GPx in the heart lysates from DOX-treated rats [1,26]. MDA showed a notable depletion with a corresponding increase in DOX concentration and/or incubation time [1]. These alterations were reversed by PEN administration, highlighting its antioxidant capacity, likely attributed to its thiol (-SH) and amino (-NH_2_) functional groups of penicillamine, which are crucial for its redox status, empowering its ROS scavenger activity [54,55]. This redox-stabilizing effect parallels PEN’s reported restoration of glutathione and catalase activity in Wilson’s disease [44].

Copper has determinative redox reactivity by generating toxic hydroxyl free radicals. It could alter the activity of the respiratory chain enzymes matrix, leading to augmented mitochondrial ROS production, oxidative stress, ultimately cellular death, and progression of numerous disorders [56,57,58]. Contemporary research identified cuproptosis, which is a copper-dependent process. The control of cuproptosis is contingent upon copper surplus, mitochondrial respiration, and ATP generation, all of which are involved in the TCA cycle. The mechanism of cuproptosis is facilitated by the direct interaction of copper ions with the lipoylated component of the TCA cycle. This results in the accumulation of lipid-acylated proteins and the downregulation of iron-sulfur cluster proteins, leading to protein toxicity stress and eventually cell death [59]. In our study, DOX upregulated FDX1, LIAS, and SLC31A1 and suppressed ATP7A, reflecting an imbalance in copper trafficking and a shift towards cuproptosis. Mechanistically, DOX-induced upregulation of FDX1 level, a key upstream regulator of protein lipoylation, triggered the accumulation of lipoylated mitochondrial protein. Concurrently, elevated LIAS expression augmented the pool of targeted lipoylated proteins, inducing proteotoxic stress. Moreover, amplified SLC31A1 expression facilitates copper influx while dampening ATP7A levels, impairing copper export. These DOX-induced alterations in cuproptosis-related gene expression provide a cellular milieu susceptible to copper-dependent cell death. These findings were corroborated by immunohistochemical analysis, where increased expression of FDX1, DLAT, and SLC31A1 in cardiac tissues was significantly attenuated by PEN treatment. PEN’s ability to normalize these markers’ expressions supports its role in inhibiting cuproptosis and reducing cardiac damage. Consistent with our results, previous studyreported that DOX treatment exacerbated myocardial cuproptosis and cardiac impairments in mice, evidenced by increased myocardial levels of DLAT accumulation and copper content in DOX-induced mice. The administration of copper chelator tetrathiomolybdate alleviated DOX-induced cardiotoxicity, indicating the involvement of cuproptosis in DOX-induced cardiac injury. Previous research also demonstrated that the intracellular Cu accumulation in cardiomyocytes of mice evoked features of cuproptosis in myocardium, including upregulated SLC31A1, DLAT, and unexpectedly downregulated FDX1 in a murine model of cardiomyocytes, possibly due to a compensatory response of intracellular Cu accumulation upon DOX challenge [60]. Collectively, PEN has significant therapeutic promise in mitigating cuproptosis and safeguarding against DOX-induced cardiac damage by successfully rectifying the DOX-induced dysregulation of cuproptosis regulators, especially correcting the expression of FDX1, LIAS, SLC31A1, DLAT, and ATP7A.

Network pharmacology analysis was employed to support these experimental findings further and delineate the molecular targets and biological processes potentially modulated by PEN. By integrating data from multiple databases, we identified 14 overlapping genes involved in DiCM and cuproptosis, including FDX1, DLAT, and SLC31A1. Enrichment analysis revealed critical pathways such as oxidative stress response, copper ion homeostasis, apoptotic regulation, and hypoxia adaptation, all known contributors to cardiomyocyte injury and survival. Notably, key signaling pathways, including renin-angiotensin system, HIF-1, apelin, and relaxin, were also significantly enriched, indicating PEN’s broad regulatory influence on cardiovascular function.

The RAS pathway is central to oxidative stress [61] and cardiac remodeling in DOX-induced cardiotoxicity, with targets such as ACE, NOS1, and NOS2 playing direct roles. PEN’s modulation of these targets corresponds with our observed reduction in myocardial fibrosis and restoration of cardiac function. Similarly, HIF-1 signaling, which is activated under mitochondrial stress and regulates cuproptosis-associated genes (EGLN1, CA9) [62], was attenuated in PEN-treated groups. This aligns with our immunostaining results showing normalization of FDX1, DLAT, and SLC31A1, suggesting that PEN mitigates hypoxia-driven copper toxicity. Enriching the Relaxin pathway, which regulates extracellular matrix turnover, explains the histologically decreased fibrosis mechanistically [63]. At the same time, Apelin signaling, a key regulator of angiogenesis and contractility, may contribute to the preserved LVEF and FS in treated animals [64]. Collectively, these pathway-level interactions reinforce a coherent mechanistic framework in which PEN’s copper-chelating and multi-targeted effects converge to suppress cuproptosis, attenuate maladaptive remodeling, and restore cardiac function.

The molecular docking results reinforce the hypothesis that penicillamine (PEN) exerts its cardioprotective effects against DiCM by directly interacting with key cuproptosis-associated proteins: FDX1, SLC31A1, and DLAT. FDX1 (Ferredoxin 1), a central upstream regulator of cuproptosis, showed a notable binding affinity with PEN, with a docking score of −3.89 kcal/mol. PEN formed multiple hydrogen bonds and salt bridges, particularly with residues Asn73, Arg74, and Asp101. This interaction is mechanistically significant given that FDX1 facilitates the reduction of Cu^2+^ to Cu^+^, directly triggering lipoylated protein aggregation in mitochondria, a hallmark of cuproptosis. A recent study has emphasized FDX1’s pivotal role in driving copper toxicity in cells via protein lipoylation and destabilization of Fe-S cluster proteins [65]. The strong interaction between PEN and FDX1 suggests that PEN may inhibit FDX1 function, thereby disrupting the copper reduction cascade and subsequent toxic stress in cardiomyocytes.

SLC31A1, a high-affinity copper importer, also demonstrated a favorable binding profile with PEN (docking score −3.53 kcal/mol), forming hydrogen bonds with residues Met19 and Leu24. Although the positive control interacted with Gly23 through hydrogen bonding, it also demonstrated binding capacity at Met19 and Leu24 via hydrophobic and van der Waals interactions, thereby confirming the reliability of the observed binding mode. According to a study published by Bian et al., SLC31A1 upregulation facilitates intracellular copper accumulation and sensitizes cells to cuproptosis. Our in vivo findings showed a significant DOX-induced upregulation of SLC31A1, which was effectively downregulated upon PEN treatment. This supports the docking data and highlights SLC31A1 as a key pharmacological target for PEN’s decoppering action [66].

DLAT (dihydrolipoamide acetyltransferase), a lipoylated component of the pyruvate dehydrogenase complex, plays a downstream role in the cuproptosis mechanism. PEN exhibited the strongest binding with DLAT among the three targets (−4.45 kcal/mol), establishing salt bridges with Arg535 and ionic interactions with Glu407. This is consistent with findings from Genes [67], where DLAT was implicated as a core protein in copper-induced proteotoxicity. Furthermore, the binding pattern of PEN closely mirrors that of lipoic acid, which occupies the same pocket and establishes comparable salt bridges and core interactions, thereby reinforcing the functional relevance of PEN binding to DLAT. Inhibition of DLAT’s interaction with Cu^+^ could prevent toxic protein aggregation and restore mitochondrial integrity—critical for cardiac cell survival under DOX stress.

The docking results validate PEN’s potential to modulate key cuproptosis-related proteins at both upstream and downstream levels. The binding interactions suggest that PEN may act as a copper chelator and a direct modulator of cuproptosis machinery, attenuating cardiomyocyte injury. These findings align well with the network pharmacology and in vivo results, collectively supporting PEN as a promising therapeutic agent in preventing DiCM via suppression of copper-mediated cell death pathways.

The overlapping protein–ligand networks and strong binding energies emphasize PEN’s multi-targeted capability in mitigating copper-mediated mitochondrial toxicity.

## 5. Conclusions

These findings compellingly demonstrate that penicillamine provides significant cardioprotection by inhibiting cuproptosis, reducing oxidative stress, and restoring cardiac function. Our multifaceted study, which skillfully combines experimental, computational, and bioinformatics methods, presents a complete and detailed picture of penicillamine’s mechanism of action and its potential as a protective agent. Given the well-documented cardiac risks of doxorubicin therapy, penicillamine stands out as a promising and practical addition to existing treatment protocols, deserving of further research in clinical and translational settings.

## Figures and Tables

**Figure 1 biomolecules-15-01320-f001:**
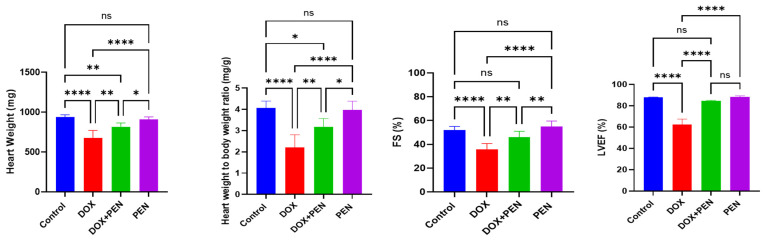
Treatment of PEN alleviates cardiac atrophy and echocardiographic parameters in an in vivo model of DOX-induced chronic cardiotoxicity. Heart weight to body weight ratio was dramatically decreased by DOX, which was ameliorated by PEN co-administration. DOX altered cardiacechocardiography parameters while PEN improved cardiac function. DOX: doxorubicin; PEN: penicillamine; LVEF: left ventricular ejection fraction; FS: Fractional shortening. * *p* < 0.05; ** *p* < 0.01; **** *p* < 0.0001, ns = non-significant.

**Figure 2 biomolecules-15-01320-f002:**
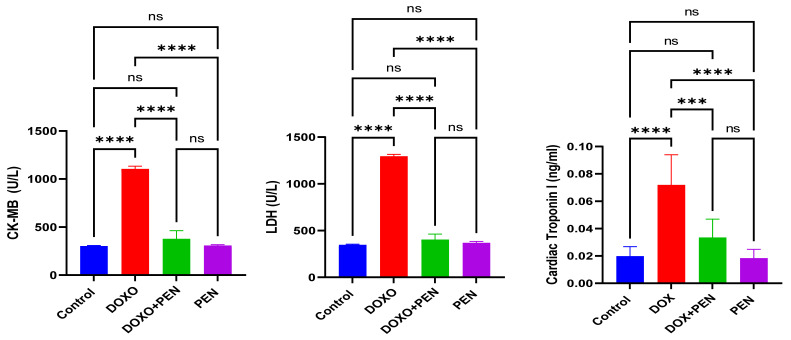
PEN treatment normalizes serum protein markers of cardiac injury (CK-MB, LDH, Cardiac troponin I) assessed by ELISA. Serum levels of cardiac markers were significantly increased by DOX, while PEN administration normalized them. DOX: Doxorubicin; PEN: Penicillamine; LDH: Lactic acid dehydrogenase; CK-MB: Creatine kinase-MB. *** *p* < 0.001, **** *p*< 0.0001, ns = non-significant.

**Figure 3 biomolecules-15-01320-f003:**
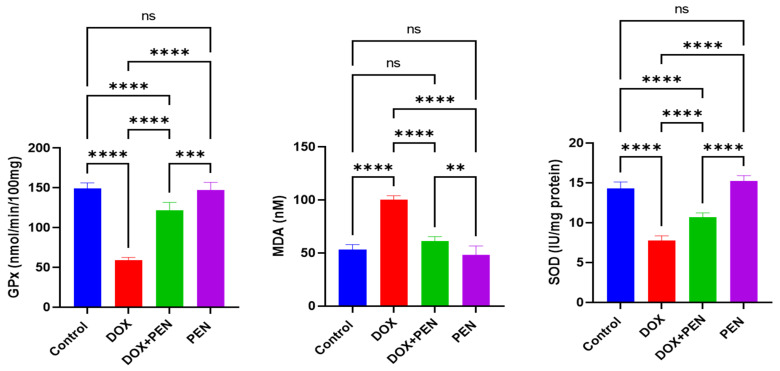
Cardiac tissue contents of oxidative stress markers (glutathione peroxidase, malondialdyde and superoxide dismutase) in DOX and PEN-treated rats measured by ELISA. The DOX treatment induced an increase in MDA level with depletion of both SOD and GPx. PEN administration reversed these changes. DOX: doxorubicin; PEN: penicillamine; MDA: malondialdehyde; SOD: superoxide dismutase, GPx: glutathione peroxidase. ** *p* < 0.01, *** *p* < 0.001, **** *p* < 0.0001, ns = non-significant.

**Figure 4 biomolecules-15-01320-f004:**
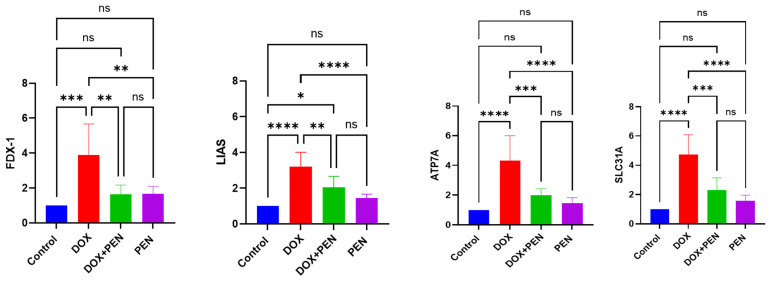
Analysis of mRNA expression levels of cuproptosis mediators-FDX-1, LIAS, ATP7A and SLC31A- in cardiac tissue of DOX and PEN-treated rats assessed by RT-PCR. The DOX treatment induced an increase in FDX-1, LIAS, and SLC31A expression and dampened ATP7A expression. PEN administration reversed these changes DOX: doxorubicin; PEN: penicillamine; FDX1: ferrordexin-1; LIAS: lipoic acid synthetase; ATP7A: ATPase copper transporting alpha; SLC31A: solute carrier family 31 member 1. * *p* < 0.05; ** *p* < 0.01; *** *p* < 0.001, **** *p* < 0.0001, ns = non-significant.

**Figure 5 biomolecules-15-01320-f005:**
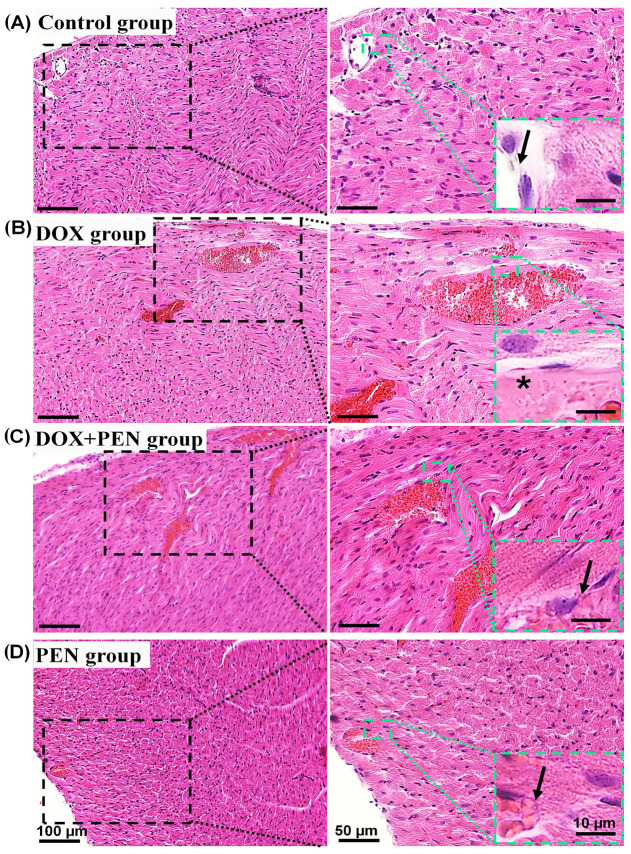
Histopathological features of HE-stained cardiac section sections among control (**A**); DOX (**B**); DOX+PEN (**C**), and PEN (**D**) groups. The photographs in the right column are higher magnifications of the dotted black squares in the left column. The green dotted insets are higher magnifications of the green dotted squares in the right column. Notice normal myocardial structure, with actin-myosin striations and thin basal lamina (arrows) surrounding the endothelial cells in control (**A**); DOX+PEN (**C**) and PEN (**D**) groups. Thick basal lamina (*), and less prominent actin-myosin striations in the DOX group (**B**). HE: Hematoxylin/Eosin; DOX: doxorubicin; PEN: penicillamine. Bars = 100; 50; and 10 for the left column; right column; and magnified images for the inset right column, respectively.

**Figure 6 biomolecules-15-01320-f006:**
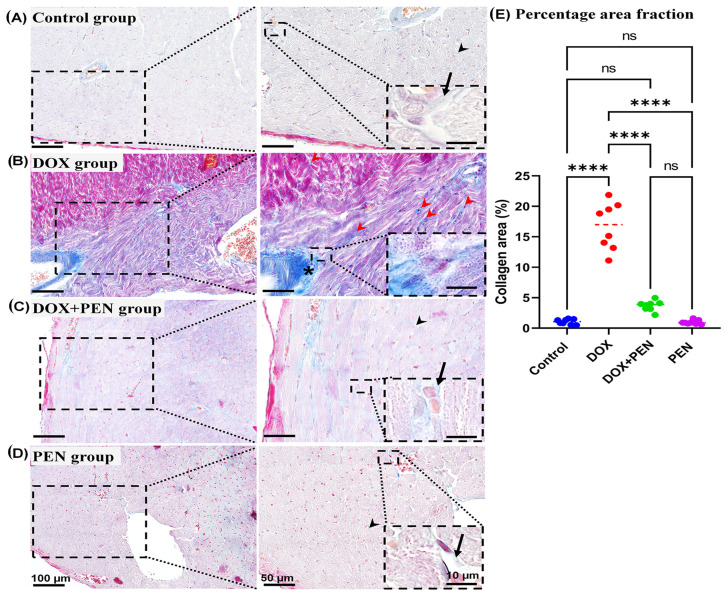
Analysis of the level of myocardial fibrosis. (**A**–**D**) Representative images of MT-stained cardiac sections among control (**A**), DOX (**B**), DOX+PEN (**C**), and PEN (**D**) groups. (**E**) Graph showing the morphometrical data of the percentage for the positive collagen fiber area fraction. The photographs in the right column are higher magnifications of the dotted black squares in the left column. The black dotted insets are higher magnifications of the black dotted squares in the right column. Notice scarce collagen fibers among the cardiac muscle (black arrow heads) and around the blood vessels (arrows) in control (**A**); DOX+PEN (**C**) and PEN (**D**) groups; Numerous collagen fibers around the blood vessels (*), and cardiac muscle (red arrow heads) in the DOX group (**B**). Graph of the statistical analysis of fibrosis level in MT-stained sections (**E**). Statistical analysis was conducted using the GraphPad Prism 9 software. Shapiro–Wilk’s test was used to test the normality of quantitative data, and one-way ANOVA was conducted for multiple comparisons among different experimental groups. (*) indicates a significant difference between studied groups: **** *p* < 0.0001, ns = non-significant. *n* = 8 in each experimental group, data are represented as mean values  ±  standard error (SE). Bars = 100; 50; and 10 for the left column; right column; and magnified images for the inset right column, respectively.

**Figure 7 biomolecules-15-01320-f007:**
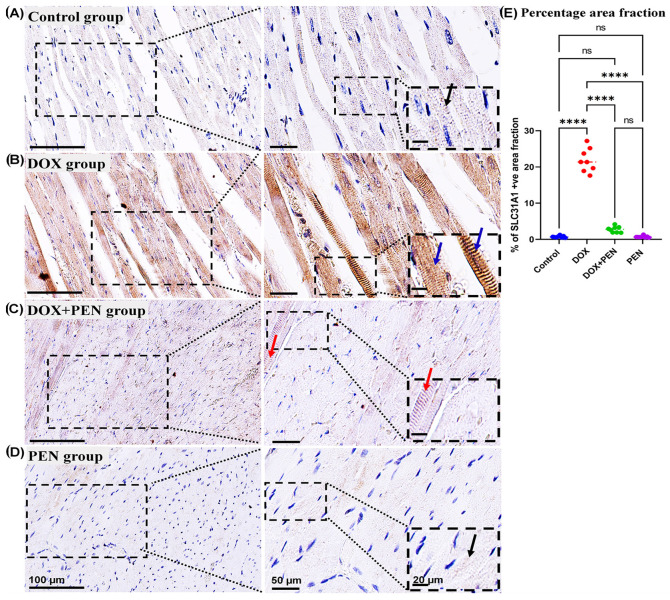
Analysis of the expression level of myocardial SLC31A1. (**A**–**D**) Representative images and (**E**) quantitative analysis of SLC31A1-immunostained cardiac sections among control (**A**), DOX (**B**), DOX+PEN (**C**), and PEN (**D**) groups. Graph showing the morphometrical quantitative data of the percentage for the positive collagen fiber area fraction (**E**). The photographs in the right column are higher magnifications of the dotted black squares in the left column. The black dotted insets are higher magnifications of the black dotted squares in the right column. Notice faint immunopositive cytoplasmic reaction (black arrows) in the control (**A**) and PEN (**D**) groups; Strong immunopositive cytoplasmic reaction (blue arrows) in the DOX group (**B**), and weak immunopositive cytoplasmic reaction (red arrows) in the DOX+PEN (**C**) groups. Graph of the statistical analysis of the percentage of positive area fraction in SLC31A1-immunostained cardiac sections (**E**). Statistical analysis was conducted using the GraphPad Prism 9 software. The Shapiro–Wilk test was used to assess the normality of the quantitative data, and one-way ANOVA was conducted for multiple comparisons among the different experimental groups. **** *p* < 0.0001, ns: non-significant. *n* = 8 in each experimental group, data are represented as mean values  ±  standard error (SE). Bars = 100; 50; and 20 for the left column; right column; and magnified images for the inset right column, respectively.

**Figure 8 biomolecules-15-01320-f008:**
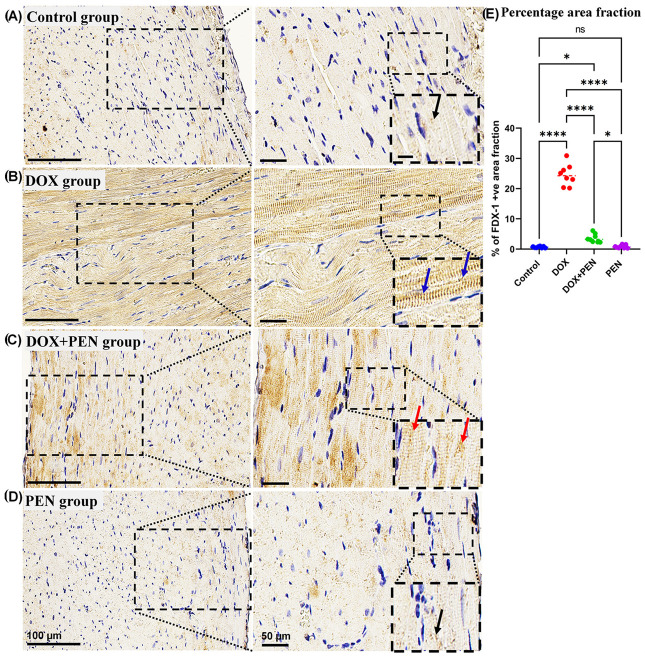
Analysis of the expression level of myocardial FDX1. (**A**–**D**) Representative images and (**E**) quantitative analysis of SLC31A1-immunostained cardiac sections among control (**A**), DOX (**B**), DOX+PEN (**C**), and PEN (**D**) groups. Graph showing the morphometrical quantitative data of the percentage for the positive collagen fiber area fraction (**E**). The photographs in the right column are higher magnifications of the dotted black squares in the left column. The black dotted insets are higher magnifications of the black dotted squares in the right column. Notice faint immunopositive cytoplasmic reaction (black arrows) in the control (**A**) and PEN (**D**) groups; Strong immunopositive cytoplasmic reaction (blue arrows) in the DOX group (**B**), and weak immunopositive cytoplasmic reaction (red arrows) in the DOX+PEN (**C**) groups. Graph of the statistical analysis of the percentage of positive area fraction in FDX1-immunostained cardiac sections (**E**). Statistical analysis was conducted using the GraphPad Prism 9 software. The Shapiro–Wilk test was used to assess the normality of the quantitative data, and one-way ANOVA was conducted for multiple comparisons among the different experimental groups. (*) indicates a significant difference between studied groups: * *p* < 0.05, **** *p* < 0.0001, ns = non-significant. *n* = 8 in each experimental group, data are represented as mean values  ±  standard error (SE). Bars = 100; 50; and 20 for the left column; right column; and magnified images for the inset right column, respectively.

**Figure 9 biomolecules-15-01320-f009:**
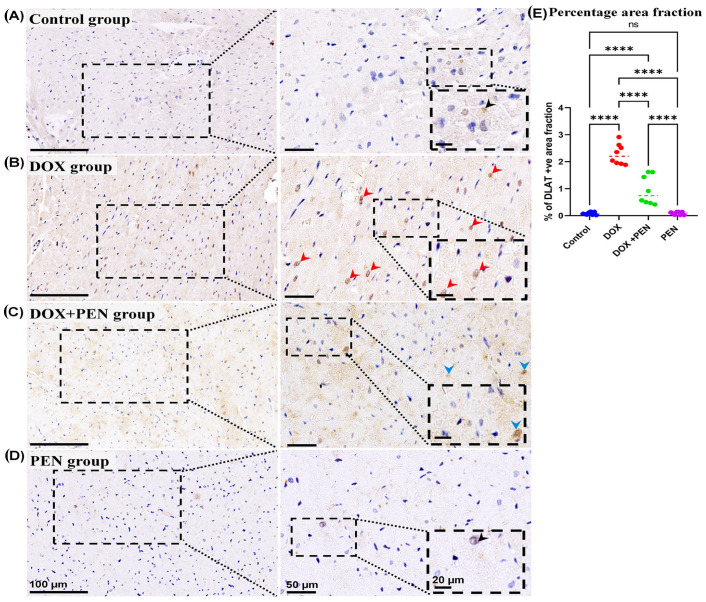
Analysis of the expression level of myocardial DLAT. (**A**–**D**) Representative images and (**E**) quantitative analysis of DLAT-immunostained cardiac sections among control (**A**), DOX (**B**), DOX+PEN (**C**), and PEN (**D**) groups. Graph showing the morphometrical quantitative data of the percentage for the positive collagen fiber area fraction (**E**). The photographs in the right column are higher magnifications of the dotted black squares in the left column. The black dotted insets are higher magnifications of the black dotted squares in the right column. Notice faint immunopositive nuclear reaction (black arrow heads) in the control (**A**) and PEN (**D**) groups; Strong immunopositive nuclear reaction (red arrow heads) in the DOX group (**B**), and weak immunopositive nuclear reaction (blue arrow heads) in the DOX+PEN (**C**) groups. Graph of the statistical analysis of the percentage of positive area fraction in SLC31A1-immunostained cardiac sections (**E**). Statistical analysis was conducted using the GraphPad Prism 9 software. The Shapiro–Wilk test was used to assess the normality of the quantitative data, and one-way ANOVA was conducted for multiple comparisons among the different experimental groups. **** *p* < 0.0001, ns: non-significant. n = 8 in each experimental group, data are represented as mean values  ±  standard error (SE). Bars = 100; 50; and 20 for the left column; right column; and magnified images for the inset right column, respectively.

**Figure 10 biomolecules-15-01320-f010:**
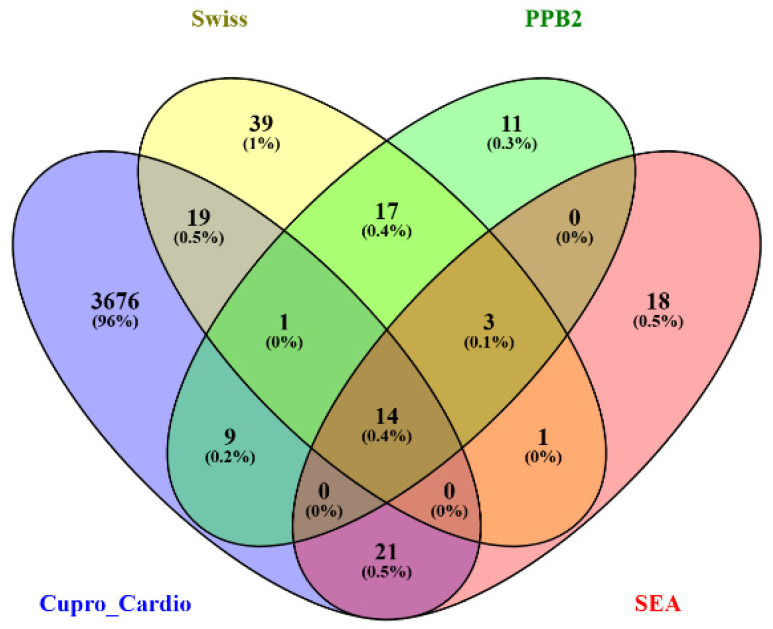
Venn diagram showing the overlap between penicillamine’s predicted targets and genes related to DiCM and cuproptosis.

**Figure 11 biomolecules-15-01320-f011:**
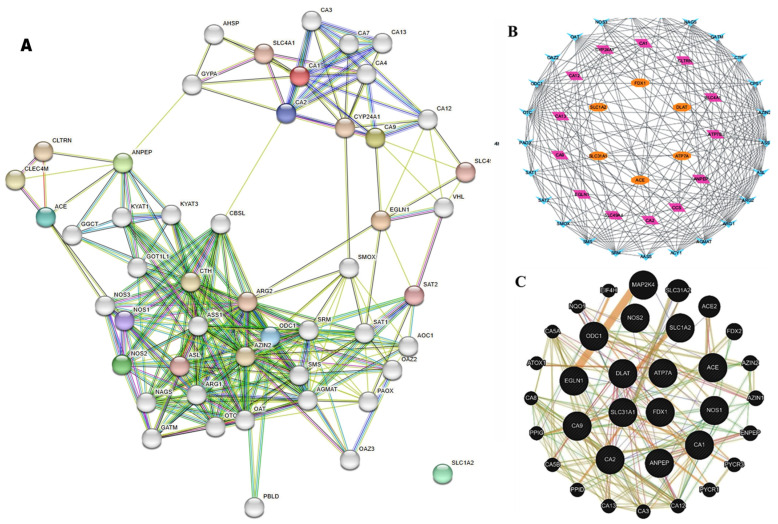
Comprehensive protein–protein interaction (PPI) network analysis of the 14 overlapping targets: (**A**) PPI network of the overlapping 14 targets constructed using the STRING database with a confidence score > 0.4, visualized in Cytoscape. (**B**) Hub gene analysis using CytoHubba reveals key regulatory nodes with high connectivity in the network. (**C**) GeneMANIA interaction network showing predicted gene functions and associations based on co-expression, physical interactions, and shared pathways.

**Figure 12 biomolecules-15-01320-f012:**
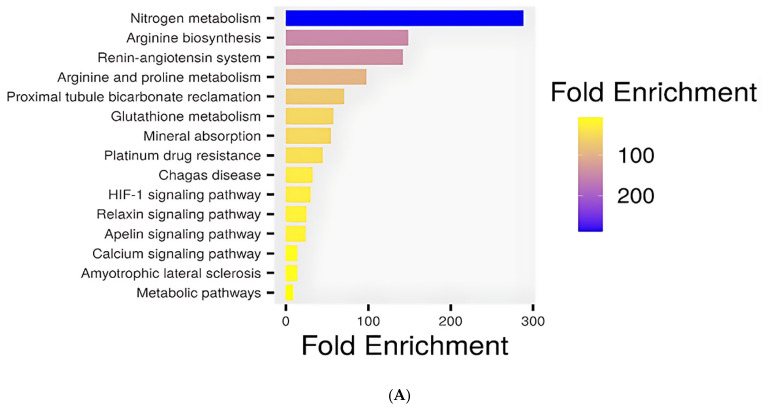
(**A**): KEGG pathway enrichment analysis of overlapping targets between penicillamine (PEN) and genes associated with DOX-induced cardiotoxicity and cuproptosis. Bar plot illustrating the top enriched KEGG pathways. (**B**): KEGG pathway enrichment analysis of overlapping targets between penicillamine (PEN) and genes associated with DOX-induced cardiotoxicity and cuproptosis enrichment of the Renin-Angiotensin system. (**C**): KEGG pathway enrichment analysis of overlapping targets between penicillamine (PEN) and genes associated with DOX-induced cardiotoxicity and cuproptosis enrichment of the HIF-1 Signaling pathway. (**D**): KEGG pathway enrichment analysis of overlapping targets between penicillamine (PEN) and genes associated with DOX-induced cardiotoxicity and cuproptosis enrichment of the Relaxin signaling pathway. (**E**): KEGG pathway enrichment analysis of overlapping targets between penicillamine (PEN) and genes associated with DOX-induced cardiotoxicity and cuproptosis enrichment of the Apelin signaling pathway.

**Figure 13 biomolecules-15-01320-f013:**
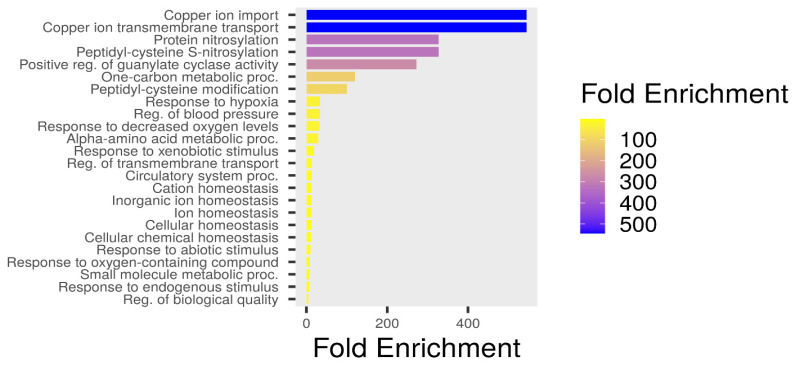
Gene Ontology Biological Process (GO: BP) enrichment analysis of the 14 overlapping targets between Penicillamine (PEN) and DOX-induced cardiotoxicity/cuproptosis-associated genes.

**Figure 14 biomolecules-15-01320-f014:**
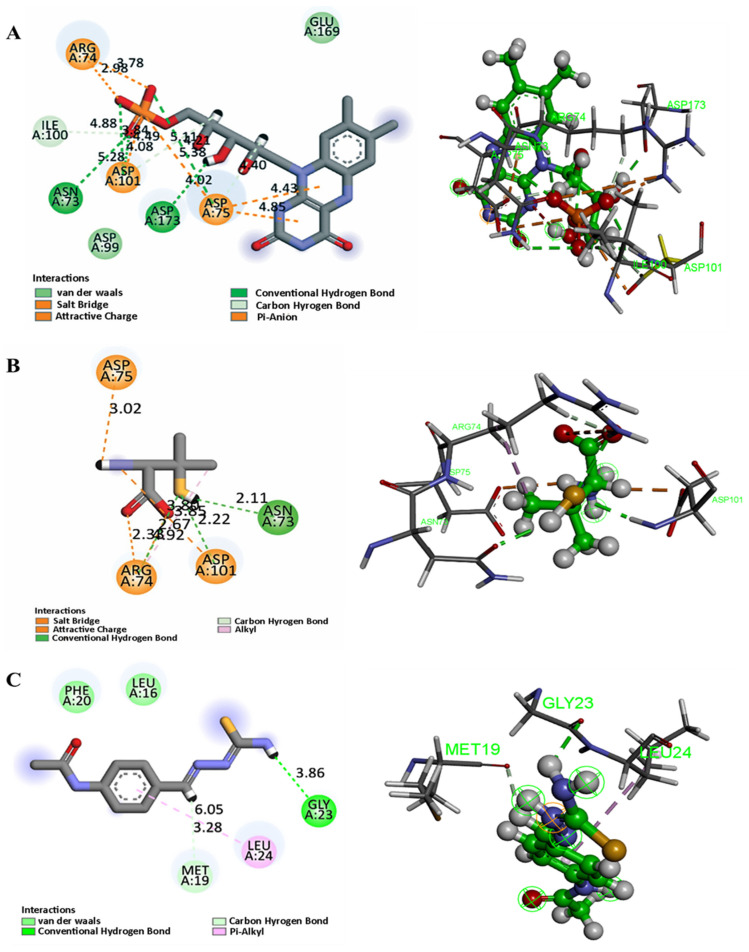
Two-dimensional and 3D interaction diagrams represent the key for the types of interaction between controls and selected protein receptors: (**A**) FDX1 (ferredoxin 1) 3P1M protein, (**C**) SLC31A13 2LS2 protein, (**E**) DLAT (dihydrolipoamide acetyltransferase) 3B8K protein. penicillamine and selected protein receptors: (**B**) FDX1 (ferredoxin 1) 3P1M protein, (**D**) SLC31A13 2LS2 protein, (**F**) DLAT (dihydrolipoamide acetyltransferase) 3B8K protein.

**Table 1 biomolecules-15-01320-t001:** Primer sequence of the tested genes.

Gene	Primer Sequence	Temperature	Accession Number	Product Size
FDX1	F:5′GGTGAAACGCTAACGACCAA-3′ R:5′-GGTAGAGCAAGCCAAAGTCC-3′	Tm = 58.6	NM_017126.2	117 bp
LIAS	F:5′-TATGTGAGGAAGCCCGATGT-3′ R:5′-CAACCTCTTGTGCATGTGTCC-3′	Tm = 59.1	NM_001012037.1	109 bp
SLC31A1	F:5′-TATTTGGTGGCTGGGGTTCT-3′ R:5′-GTGCACTAGGTCTGGAGAGG-3′	Tm = 59.1	NM_133600.3	154 bp
ATP7A	F:5′TAGACGGCATGCATTGTAAATC-3′ R:5′-TGGATTTTACACCTGGCTTCTT-3′	Tm = 57.5	NM_052803.2	375 bp
GAPDH	F:5′-GAGACAGCCGCATCTTCTTG-3′ R:5′-TGACTGTGCCGTTGAACTTG-3′	Tm = 58.9	NM_017008.4	224 bp

**Table 2 biomolecules-15-01320-t002:** Antibodies list, source/catalog number, dilutions, and antigen retrieval conditions.

Antibody	Source and Catalog Number	Working Dilution	Antigen Retrieval
Rabbit anti-SLC31A1/CTR1	GeneTex (Cat. No. GTX48534)	1:300	Tris-HCl buffer (pH 9)/110 °C, 15 min
Rabbit anti-FDX1	Proteintech (Cat. No. 12592-1-AP)	1:100	Tris-HCl buffer (pH 9)/110 °C, 15 min
Rabbit anti-DLAT	CUSABIO (Cat. No.CSB-PA445587)	1:50	Tris-HCl buffer (pH 9)/110 °C, 15 min

**Table 3 biomolecules-15-01320-t003:** Best Docking Pose and Key Interactions of Penicillamine with Cuproptosis-Associated Targets.

Protein (PDB ID)		Docking Score	Key Interacting Residues	Interaction Type	Distance (Å)
FDX1 (3P1M)	Control	−5.16	Asn73	H-bond	5.28
			Arg74	Salt-bridge	2.98 3.78
			Asp75	2Pi-anion + Ionic bond	4.43 4.85 5.38
			Asp101	H-bond + Ionic bond	4.88 4.08
			Asp173	H-bond	4.02
	PEN	−3.89	Asn73	H-bond	2.11
			Arg74	Salt-bridge	2.33 1.92
				H-bond	2.67
			Asp75	Ionic bond	3.02
			Asp101	Ionic bond	2.22
				H-bond	3.80
SLC31A1 (2LS2)	Control	−3.97	Gly23	H-bond	3.86
	PEN	−3.53	Met19	H-bond	3.02
				H-bond	2.01
			Leu24	H-bond	3.05
DLAT (3B8K)	Control	−5.52	Asp533	H-bond	3.23
			Arg535	Salt-bridge	6.03 7.72
			Trp545	Pi-sulfur	4.93
	PEN	−4.45	Arg535	Salt-bridge	3.20 5.15
			Glu407	Ionic bond	2.30

## Data Availability

The original contributions presented in this study are included in the article. Further inquiries can be directed to the corresponding author(s).

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
