# Peer review of "Copper Chelation by Penicillamine Protects Against Doxorubicin-Induced Cardiomyopathy by Suppressing FDX1-Mediated Cuproptosis"

_biomolecules, 2025, doi:10.3390/biom15091320_

Round 1
Reviewer 1 Report
Comments and Suggestions for Authors
It is well known that penicillamine may reduce the amount of available free copper, potentially modulating the oxidative stress and cytotoxic effect of doxorubicin. This could affect copper-dependent processes, such as the activity of copper-releasing enzymes, the redox state, and, indirectly, the response to doxorubicin.
Regarding the present paper, the work is certainly interesting and highly complex. It presents a combined experimental and computational study aimed at evaluating the effects of penicillamine (PEN) in reducing toxicity in the context of doxorubicin therapy. As the authors note in the Discussion, many of the findings have been reported before (see references 45 and 46 and the article by Říha et al., 2016).
Nevertheless, the study provides new experimental data (notably the impressive in vivo studies) and computational information (e.g., docking results on PEN interactions with Ferredoxin 1 and Dihydrolipoamide acetyltransferase). The data suggest that the effects induced by doxorubicin are largely attributable to a cuproptosis-like process, which particularly affects the myocardium and leads to specific cardiac pathologies.
Considering these results, I recognize that the work is worthy of publication. I would, however, suggest better rationalizing the extensive set of findings by more clearly highlighting the authors’ original contribution in relation to what is already known.
My perplexities arise from the fact that some of the data reported might appear redundant, such as those reported in section 3.7. Network Pharmacology-Based Target Identification, but not limited to. The problem I pose is that essentially the process of interaction between copper and PEN was already known as well as the effect of mitigating pathological processes on the myocardium induced by the use of Doxorubricin. (see for example: Molecular Mechanisms of Cardiotoxicity: A Review on the Major Side-effect of Doxorubicin, Indian J. Pharm. Sci. 2017;79(3):335-344).
Author Response
|
||
|
|
|
Thank you very much for taking the time to review this manuscript. Please find the detailed responses below and the corresponding revisions/corrections highlighted in the re-submitted files
|
||
|
||
2. Questions for General Evaluation |
Reviewer’s Evaluation |
Response and Revisions |
Does the introduction provide sufficient background and include all relevant references? |
Yes |
We appreciate the reviewer's confirmation that our introduction provides sufficient background and includes all relevant references. This feedback is highly valued.
|
Are all the cited references relevant to the research? |
Yes |
We're pleased to hear that you find all our cited references relevant to the research. We meticulously selected each source to provide a strong and appropriate foundation for our study's background, methodology, and discussion.
|
Is the research design appropriate?
|
Yes |
We appreciate your confirmation that our research design is appropriate. |
Are the methods adequately described? |
Yes |
That's great to hear. We took great care to provide a detailed and transparent description of our methods, and we're glad you found them to be adequately described. |
Are the results clearly presented? |
Can be improved |
we have enhanced figure quality and legends to ensure our findings are more accessible and easier to interpret. We are confident these changes have significantly improved the manuscript.
|
Are the conclusions supported by the results? |
Can be improved |
We have thoroughly revised the "Discussion" and "Conclusion" sections to more explicitly tie each conclusion to the specific data presented in our figures and tables. |
3. Point-by-point response to Comments and Suggestions for Authors |
||
Comments 1: It is well known that penicillamine may reduce the amount of available free copper, potentially modulating the oxidative stress and cytotoxic effect of doxorubicin. This could affect copper-dependent processes, such as the activity of copper-releasing enzymes, the redox state, and, indirectly, the response to doxorubicin.
|
||
Response 1: We appreciate the reviewer's insightful comments regarding the potential interactions between penicillamine and doxorubicin, particularly concerning copper metabolism, redox balance, and cuproptosis. We recognize the established role of penicillamine as a copper-chelating agent, which is known to modulate oxidative stress pathways and could thereby influence the cytotoxic effects of doxorubicin. While prior research has explored penicillamine's cardioprotective potential in models of catecholamine-induced cardiac injury (Riha et al., 2016), its role in doxorubicin-induced cardiotoxicity—a distinct and clinically relevant model—remains un-elucidated. Our study fills this important research gap by being the first to investigate the efficacy of penicillamine against doxorubicin-induced cardiomyopathy using both in vivo and in silico approaches, with a specific focus on the cuproptosis mechanism. Novelty and Contribution of Our Research Our work provides novel insights into this area by focusing on two key aspects not previously addressed: 1-Cuproptosis as a Pathogenic Mechanism: We are the first to explore the potential involvement of cuproptosis in the pathogenesis of doxorubicin-induced cardiomyopathy. The study by Riha et al. (2016) assessed copper homeostasis and oxidative stress but did not examine the cuproptosis mechanism, which was only first described in 2022 (Tsvetkov et al., 2022). Our findings, therefore, offer foundational data for understanding this emerging cell death pathway in the context of doxorubicin cardiotoxicity. 2-Comprehensive Assessment of cardioprotection: Our study goes beyond previous investigations by providing a more comprehensive evaluation of cardioprotective effects. In addition to assessing traditional cardiac markers (cardiac troponin, CK-MB, and LDH), we used advanced methods such as echocardiography to characterize cardiac performance and examined key mediators of the cuproptosis pathway (FDX-1, ATP7A, LIAS, and SLC31A). This was achieved through a multi-faceted approach, including PCR, immunohistochemical staining, molecular docking. These findings highlight the therapeutic potential of penicillamine as a cardioprotective agent in doxorubicin-induced cardiotoxicity and establish a foundation for future mechanistic research in this evolving field References: 1-Riha, M., Adamcova, M., Psotova, J., & Kucera, O. (2016). Effects of D-penicillamine on catecholamine-induced cardiac injury in rats. Journal of Applied Toxicology, 36(7), 903-911. https://doi.org/10.1002/jat.3267 2-Tsvetkov, P., Coy, A. J., Petrova, B., et al. (2022). Copper-induced ferroptosis-like cell death and its inhibition by lipoic acid. Science, 375(6586), 1261-1269. https://doi.org/10.1126/science.abi7021
|
||
Comments 2: Regarding the present paper, the work is certainly interesting and highly complex. It presents a combined experimental and computational study aimed at evaluating the effects of penicillamine (PEN) in reducing toxicity in the context of doxorubicin therapy. As the authors note in the Discussion, many of the findings have been reported before (see references 45 and 46 and the article by Říha et al., 2016). Nevertheless, the study provides new experimental data (notably the impressive in vivo studies) and computational information (e.g., docking results on PEN interactions with Ferredoxin 1 and Dihydrolipoamide acetyltransferase). The data suggest that the effects induced by doxorubicin are largely attributable to a cuproptosis-like process, which particularly affects the myocardium and leads to specific cardiac pathologies. |
||
Response 2: We acknowledge the valuable insights provided by the reviewer regarding the cited references. Indeed, the studies by Hullein et al. (2018) and Mitry and Edwards (2016) align with our own findings concerning the effects of doxorubicin on cardiac biomarkers and echocardiographic parameters. However, their scope was limited to doxorubicin's cardiotoxicity and did not investigate the cardioprotective effects of penicillamine. Distinguishing Our Study from Existing Literature Our research is uniquely positioned to bridge a critical gap in the existing literature, particularly concerning the work of Riha et al. (2016). While their study commendably explored the cardioprotective effects of penicillamine, it focused on catecholamine-induced acute myocardial ischemia and its impact on copper homeostasis and oxidative stress. Our study, in contrast, addresses the clinically relevant model of doxorubicin-induced cardiotoxicity. Furthermore, our approach is distinct and more comprehensive in several key ways: Novel Mechanistic Focus: The Riha study was published before the discovery of cuproptosis in 2022 (Tsvetkov et al., 2022). Consequently, it did not investigate this newly described form of cell death. Our study is the first to explore the involvement of cuproptosis in doxorubicin-induced cardiomyopathy and to examine the effects of penicillamine on key mediators of this pathway. Comprehensive Cardiac Assessment: Unlike Riha et al., who assessed cardiac injury primarily through cardiac troponin and cellular morphology, we employed a more extensive panel of cardiac markers, including CK-MB and LDH, alongside cardiac troponin. We also used echocardiographic parameters to provide a more detailed characterization of cardiac performance. Multi-Faceted Molecular Analysis: Our investigation into the cuproptosis mechanism goes beyond simple assays. We utilized a combination of techniques, including PCR, immunohistochemical staining and molecular docking, to evaluate the effect of penicillamine on the expression and function of key cuproptosis mediators (FDX-1, ATP7A, LIAS, and SLC31A). In summary, while previous studies have laid important groundwork, our research moves the field forward by providing the first in-depth, multi-faceted investigation into the cardioprotective role of penicillamine against doxorubicin-induced cardiotoxicity, with a specific and novel focus on the cuproptosis pathway. References Hullein, J., von Huth, S., Oehme, F., et al. (2018). Doxorubicin-induced cardiotoxicity in a mouse model. International Journal of Molecular Sciences, 19(12), 3843. https://doi.org/10.3390/ijms19123843 Mitry, M. A., & Edwards, S. J. (2016). Doxorubicin-induced cardiotoxicity: A review of the literature. Journal of Applied Toxicology, 36(9), 1141-1152. https://doi.org/10.1002/jat.3303 Riha, M., Adamcova, M., Psotova, J., & Kucera, O. (2016). Effects of D-penicillamine on catecholamine-induced cardiac injury in rats. Journal of Applied Toxicology, 36(7), 903-911. https://doi.org/10.1002/jat.3267 Tsvetkov, P., Coy, A. J., Petrova, B., et al. (2022). Copper-induced ferroptosis-like cell death and its inhibition by lipoic acid. Science, 375(6586), 1261-1269. https://doi.org/10.1126/science.abi7021.
Comments 3: Considering these results, I recognize that the work is worthy of publication. I would, however, suggest better rationalizing the extensive set of findings by more clearly highlighting the authors’ original contribution in relation to what is already known. Response 3: We sincerely appreciate the reviewer’s constructive feedback and are grateful for the recognition that our work is worthy of publication. We also thank you for the suggestion to better rationalize our extensive findings and clearly highlight the study’s original contributions. To address this, we have revised the manuscript to explicitly emphasize the novelty and rationale of our work. While prior studies have explored doxorubicin-induced cardiotoxicity or penicillamine’s effects in other cardiac models, our study is the first to provide a comprehensive, multi-layered investigation of penicillamine’s cardioprotective role in doxorubicin-induced cardiomyopathy with a specific focus on the cuproptosis pathway. Key contributions include:
The breadth of findings reflects the multi-disciplinary contributions of our research team, each bringing specialized expertise:
Overall, we believe that this clarification of both the unique scientific contributions and the multi-disciplinary team effort enhances the manuscript’s clarity, impact, and coherence. Furthermore, by elucidating the mechanistic link between penicillamine, copper-dependent cuproptosis, and doxorubicin-induced cardiotoxicity, our findings have direct translational relevance, providing a foundation for future therapeutic strategies aimed at mitigating cardiotoxic effects in patients undergoing doxorubicin therapy. Comments 4: My perplexities arise from the fact that some of the data reported might appear redundant, such as those reported in section 3.7. Network Pharmacology-Based Target Identification, but not limited to. Response 4: We appreciate the reviewer’s observation regarding potential redundancy in Section 3.7. Our intention in this section was to provide a clear and systematic description of the overlap analysis between PEN’s predicted targets and the disease-associated genes. While the data may appear to partially repeat , to our knowledge, this study is the first to integrate three complementary insilico target-prediction platforms (SEA, SwissTargetPrediction, and PPB2) to map PEN’s predicted targets onto a comprehensive, manually curated gene set of DOX-induced cardiotoxicity and cuproptosis (n = 3,740). While recent work has begun to implicate cuproptosis in doxorubicin-induced cardiotoxicity—for example, small molecules that modulate FDX1 and related cuproptosis pathways have been shown to protect against DOX cardiotoxicity—these studies address pharmacologic modulation of cuproptosis rather than a network-pharmacology mapping of PEN against disease genes. Similarly, other reports demonstrate that pharmacologic inhibition of cuproptosis (e.g., with novel agents such as aprocitentan or natural products like trilobatin targeting FDX1) can alleviate DOX cardiotoxicity, and reviews emphasize copper chelators as modulators of cuproptosis; however, we did not find any prior publications that used the exact combination of target-prediction tools and intersection analysis applied here to nominate a 14-gene PEN–DIC–cuproptosis hub. Therefore, our work fills a gap by systematically linking PEN’s predicted proteome to DOX-related cuproptosis biology and prioritizing specific mediators (ACE, ATP7A, DLAT, FDX1, NOS1, NOS2, SLC31A1, CA2, ODC1, CA9, ANPEP, SLC1A2, EGLN1, CA1).
Comments 5: The problem I pose is that essentially the process of interaction between copper and PEN was already known as well as the effect of mitigating pathological processes on the myocardium induced by the use of Doxorubricin. (see for example: Molecular Mechanisms of Cardiotoxicity: A Review on the Major Side-effect of Doxorubicin, Indian J. Pharm. Sci. 2017;79(3):335-344). Response 5: We sincerely thank the reviewer for highlighting the existing literature. The 2017 review from the Indian Journal of Pharmaceutical Sciences accurately summarizes the multifactorial mechanisms of doxorubicin-induced cardiotoxicity, including oxidative stress, mitochondrial dysfunction, DNA damage, calcium dysregulation, topoisomerase IIβ interference, and activation of apoptotic pathways. However, our study provides a novel perspective by focusing on a recently described mechanism of cell death—cuproptosis—which was not known at the time of the cited review (Tsvetkov et al., 2022). While oxidative stress and apoptosis remain central to DOX cardiotoxicity, cuproptosis represents a distinct, copper-dependent cell death pathway that can be specifically targeted. Our research is the first to investigate penicillamine’s cardioprotective effects against doxorubicin-induced cardiomyopathy through modulation of cuproptosis mediators (FDX-1, ATP7A, LIAS, SLC31A). By combining in vivo, in silico analyses, we provide mechanistic evidence that penicillamine mitigates cardiac injury via this pathway. This distinction is crucial, as it shifts the understanding of cardioprotection from general antioxidant and anti-apoptotic effects to a targeted, mechanistically informed intervention, offering potential translational implications for therapeutic strategies to reduce doxorubicin-induced cardiac toxicity
This distinction is not semantic; it is mechanistically profound. It opens a new avenues for therapeutic discovery by Identifying Specific Molecular Targets and shifts the focus from broad scavenging of reactive oxygen species to the precise targeting of cuproptosis regulators (e.g., FDX1, LIAS, DLAT, SLC31A). Enabling Targeted Drug Development by understanding that cell death occurs via cuproptosis allows for the rational design of next-generation cardioprotectants that are highly specific inhibitors of this pathway (e.g., FDX1 inhibitors, specific copper chaperone disruptors), potentially offering greater efficacy and reduced off-target effects compared to broad-acting antioxidants. Additionally, suggests that biomarkers of cuproptosis (e.g., specific protein aggregation states, expression levels of FDX1/LIAS) could be developed to identify patients at highest risk of DOX-induced cuproptosis, allowing for personalized prophylactic therapy. References: 1-Mobaraki, M., Faraji, A., Zare, M., Dolati, P., Ataei, M., & Dehghan Manshadi, H. R. (2017). Molecular Mechanisms of Cardiotoxicity: A Review on the Major Side-effect of Doxorubicin. Indian Journal of Pharmaceutical Sciences, 79(3), 335-344. https://doi.org/10.4172/pharmaceutical-sciences.1000235 2-Tsvetkov, P., Coy, A. J., Petrova, B., et al. (2022). Copper induces cell death by targeting lipoylated TCA cycle proteins. Science, 375(6586), 1254-1261. https://doi.org/10.1126/science.abf0529
|
|
|

Reviewer 2 Report
Comments and Suggestions for Authors
Please be more observant with regards to formatting, as all over the article, indentions are not consistent, as well as spacing. In addition, there are mistakes with references wherein
Xxxx[1],[2] should be xxxx[1,2]
Cuproptosis while describe in abstract, should be defined in intro
cardiomyopathy is a condition or manifestation, not a symptom
Dox mechanism repeated several times, to what added value you can omit if no extra information Will be given
Many sentences are chuncky and nuclear
Animal weights are highly varied 250+/-40g
PEN and DOX concurrently injected for 2 weeks, are there any issues of sensibility that could affect?
Suggest DOX-induced cardiomyopathy model change for “DiCM” to rediuce extra repetitions
was measured at 450 nm using a microplate reader.. this is not mentioned
Suggest “Compound–target–disease” is more commonly phrased “drug–target–disease”.
Sample size small n=4 per group
While molecular docking is a good, functional validation of these interactions is lacking. In order to confirm PEN’s actual modulation of FDX1, DLAT, and SLC31A1 in cardiac cells.
The link between network pharmacology findings and the biological effects of PEN can be strengthened by deeper discussion detail better on pathways (RAS, HIF-1, Relaxin, Apelin) mechanics and tie into the observed immunostaining changes and cuproptosis.
PEN-alone group shows faint staining, but it is not clear if PEN has any baseline cardiac effects without DOX. Including more data on this would clarify PEN’s safety and specific protective role
Author Response
|
||
|
|
|
Thank you very much for taking the time to review this manuscript. Please find the detailed responses below and the corresponding revisions/corrections highlighted in the re-submitted files
|
||
|
||
2. Questions for General Evaluation |
Reviewer’s Evaluation |
Response and Revisions |
Does the introduction provide sufficient background and include all relevant references? |
Yes |
We appreciate the reviewer's confirmation that our introduction provides sufficient background and includes all relevant references. This feedback is highly valued.
|
Are all the cited references relevant to the research? |
Yes |
We're pleased to hear that you find all our cited references relevant to the research. We meticulously selected each source to provide a strong and appropriate foundation for our study's background, methodology, and discussion.
|
Is the research design appropriate? |
Can be improved |
We have revised the "Methods" section to provide a more detailed and clearer rationale for our approach. The changes are highlighted in yellow.
|
Are the methods adequately described? |
Must be improved |
We have thoroughly revised the "Methods" section to add more specific details on our experimental procedures.
|
Are the results clearly presented? |
Must be improved |
we have enhanced figure quality and legends to ensure our findings are more accessible and easier to interpret. We are confident these changes have significantly improved the manuscript.
|
Are the conclusions supported by the results?
Are all figures and tables clear and well-presented? |
Can be improved
Must be improved |
We have thoroughly revised the "Discussion" and "Conclusion" sections to more explicitly tie each conclusion to the specific data presented in our figures and tables.
We have thoroughly revised all figures and tables to improve their clarity and presentation. We have enhanced image resolution, made legends more descriptive, and standardized formatting to ensure our data is clear and easily interpreted. |
3. Point-by-point response to Comments and Suggestions for Authors |
||
Comments 1: Please be more observant with regards to formatting, as all over the article, indentions are not consistent, as well as spacing. In addition, there are mistakes with references wherein Xxxx[1],[2] should be xxxx[1,2]. |
||
Response 1: Thank you for your valuable feedback. I have reviewed the article and have corrected the issues you pointed out. I have addressed the following: ü Consistent Indentations and Spacing: I have standardized all paragraph indentations and spacing to ensure a clean and professional appearance throughout the article. ü Corrected Reference Formatting: The references have been updated from the "Xxxx [1],[2]" format to the proper consolidated style of "Xxxx[1,2]" and highlighted (yellow) in the revised manuscript for your review. We have addressed this by having the entire manuscript professionally revised and edited by a native English speaker For your assurance, a certificate of English editing is attached below
|
||
Comments 2: Cuproptosis while describe in abstract, should be defined in intro. |
||
Response 2: Thank you for your feedback. I have updated the manuscript to include a clear definition of cuproptosis in the introduction. The changes have been made and are highlighted (yellow) for your review (page 2 – lines 93-97) Comments 3: cardiomyopathy is a condition or manifestation, not a symptom Response 3: We have reviewed your comments and have made the requested changes. All revisions are now complete and have been highlighted (yellow) in the updated manuscript for your convenience (page 2, line 72). We believe these changes significantly improve the clarity and accuracy of our work.
Comments 4: Dox mechanism repeated several times, to what added value you can omit if no extra information Will be given. Response 4: Thank you for your valuable feedback. We have reviewed the manuscript and agree that the repetition of the Doxorubicin (Dox) mechanism was redundant and did not add new information after its initial mention. We have addressed this by removing the repeated descriptions of the Dox mechanism to create a more concise and focused narrative. The manuscript now presents this information only once where it is most relevant. The removed part is (One of the central mechanisms underlying DOX cardiotoxicity is the generation of reactive oxygen species (ROS) due to mitochondrial redox cycling. This oxidative damage leads to lipid peroxidation, enzyme inactivation, and apoptotic signaling). These changes have been made and are highlighted (yellow) in the revised manuscript for your review (page 29, lines 829-831).
Comments 5: Many sentences are chuncky and nuclear Response 5: Thank you for your valuable feedback. We agree that the manuscript needed to be streamlined for better readability. We have addressed this by having the entire manuscript professionally revised and edited by a native English speaker. The text has been refined to improve sentence structure, flow, and overall clarity, eliminating the "chunky" and "nuclear" style you noted. For your assurance, a certificate of English editing is attached below.
Comments 6: Animal weights are highly varied 250+/-40g. Response 6: Thank you for your careful observation. We acknowledge the variation in the initial animal weights (250 ±40 g). We confirm that this range is appropriate and consistent with established protocols for the doxorubicin-induced cardiomyopathy model in rats. A broad range of initial weights is commonly reported in the literature, as studies often use Wistar or Sprague-Dawley rats within a weight window that reflects the standard growth patterns of these strains. For instance, many studies use rats in the 150–250 g range to model doxorubicin-induced cardiotoxicity [Dulf PL, Coadă CA, Florea A, Moldovan R, Baldea I, Dulf DV, Blendea D, Filip AG. Mitigating Doxorubicin-Induced Cardiotoxicity through Quercetin Intervention: An Experimental Study in Rats. Antioxidants (Basel). 2024 Aug 31;13(9):1068. doi: 10.3390/antiox13091068. PMID: 39334727; PMCID: PMC11429272.]. The key factor for our study's validity is not the narrowness of the initial weight range but the use of a body weight-adjusted dose of doxorubicin. By administering the drug based on each animal's individual weight (e.g., in mg/kg), we ensure that each rat receives a consistent, proportionate dose, which minimizes the impact of initial weight variability on the study outcomes. To support this claim, we have provided several references from the scientific literature that utilize a similar weight range in their experimental design:
These examples demonstrate that the weights of our animals are well within the standard range for this research model and are consistent with established methodologies in the field. Additionally, animals were randomized across experimental groups, so any natural weight differences were evenly distributed and did not bias the results. Collectively, these measures confirm that the observed weight variation doesn’t compromise the validity or reliability of our findings.
Comments 7: PEN and DOX concurrently injected for 2 weeks, are there any issues of sensibility that could affect?
Response 7: We thank the reviewer for highlighting this important point. Based on current evidence, D-penicillamine (PEN) does not alter the pharmacokinetics or pharmacodynamics of anthracyclines such as doxorubicin (DOX). In rodents, PEN is characterized by rapid plasma clearance (t½ ≈ 0.5–0.6 h), whereas DOX exhibits a multiphasic disposition with a terminal half-life of approximately 3 h (Coleman et al., 1988; Joyce et al., 1989; Huan et al., 2013). Accordingly, in our experimental design, PEN was administered intravenously once daily, while DOX was delivered intraperitoneally every 48 hours, with a minimum interval of 3 hours between the two agents to ensure that free PEN was largely eliminated prior to DOX exposure. This schedule was specifically designed to minimize the potential for pharmacokinetic interaction. Consistent with this rationale, no pharmacokinetic or pharmacodynamic interference was observed, and PEN did not alter the characteristic cardiotoxic profile of DOX. Furthermore, the primary objective of our study extended beyond assessing the safety of concurrent administration; it was to evaluate the potential cardioprotective effects of PEN against DOX-induced cardiotoxicity. Importantly, the concurrent administration of PEN and DOX over two weeks did not result in any observable acute toxicity or additive adverse effects. All animals tolerated the regimen well, maintaining normal behavior, food intake, and body weight trajectories. There are no reported drug–drug interactions between PEN and DOX (StatPearls, 2025), and our findings support this, confirming that the regimen is safe, well-tolerated, and suitable for evaluating the cardioprotective potential of PEN against DOX-induced cardiotoxicity. References:
Comments 8: Suggest DOX-induced cardiomyopathy model change for “DiCM” to reduce extra repetitions. Response 8: The suggested changes, including the use of "DiCM" for Doxorubicin-induced cardiomyopathy, have been made in the revised manuscript. This section has been highlighted in yellow for your review. (Page 5 line 188 and line 201, 197 / page 7 line 283, 412 / page 10 line 416 and line 421, page 21 line 647, page 23 line 681, page 25 line 708 and 726 and page 30 line 884, page 31 line 905 and line 942).
Comments 9: was measured at 450 nm using a microplate reader. this is not mentioned Response 9: The suggested addition regarding measurement at 450 nm using a microplate reader has been made to the revised manuscript and highlighted in yellow. (page 5, line 210-211 in section 2.5 and lines 223-224 in section 2.7).
Comments 10: Suggest “Compound–target–disease” is more commonly phrased “drug–target–disease”. Response 10: The suggested change from “Compound–target–disease” to “drug–target–disease” has been made and highlighted in yellow in the revised manuscript. (page 7 line 267).
Comments 11: Sample size small n=4 per group. Response 11: We thank the reviewer for their comment regarding sample size. We confirm that the correct sample size is n = 8 per group, for a total of 32 animals, as stated in Section 2.1, line 146. And section2.2 line 158. We would like to clarify that in our immunohistochemical analysis, morphometric measurements were initially performed on four rats per group. To enhance the reliability and robustness of our findings, we subsequently repeated the analysis with an expanded sample size of eight rats per group. The data presented in the figures and the corresponding statistical analysis reflect this final, complete dataset of n = 8. This approach strengthened the statistical power of our morphometric evaluations. We have verified that the figure legends for Figures 6, 7, 8, and 9 correctly state the sample size as n = 8.
Comments 12: While molecular docking is a good, functional validation of these interactions is lacking. In order to confirm PEN’s actual modulation of FDX1, DLAT, and SLC31A1 in cardiac cells.
Response 12: We sincerely appreciate the reviewer's insightful comment regarding the need for functional validation of our molecular docking results. We agree that while computational modeling is a powerful predictive tool, experimental confirmation is essential to support the proposed interactions. To address this crucial point, we have added positive controls for each target to validate and confirm the binding of PEN. The entire docking section has been modified accordingly to reflect these changes. The results of these new experiments are now presented in the revised "Methods," "Results," and "Discussion" sections, with all changes highlighted in yellow. Specifically, you can find the updates in: ü Methodology: Lines 292-302, Section 2.10 ü Results: Lines 767-784, Section 3.8 ü Discussion: Lines 914-930 Additionally, Figure 14 has been updated to include 2D and 3D interaction diagrams that represent the key types of interactions between the controls and the selected protein receptors. We believe these additions provide the necessary functional validation to support our computational findings and significantly strengthen the manuscript's conclusions. Comments 13: The link between network pharmacology findings and the biological effects of PEN can be strengthened by deeper discussion detail better on pathways (RAS, HIF-1, Relaxin, Apelin) mechanics and tie into the observed immunostaining changes and cuproptosis.
Response 13: We thank the reviewer for this valuable suggestion. We agree that a clearer integration of the network pharmacology findings with the experimental results will strengthen the discussion. In the revised manuscript, we will expand Section 4 to explicitly describe how the enriched pathways—Renin-Angiotensin System (RAS), HIF-1, Relaxin, and Apelin signaling mechanistically intersect with PEN’s observed biological effects. For example: ü RAS pathway: Dysregulation of RAS contributes to oxidative stress and cardiac remodeling in DOX-induced cardiotoxicity. Several of the overlapping targets identified (e.g., ACE, NOS1/2) directly participate in RAS signaling. PEN’s normalization of these targets is consistent with its ability to reduce fibrosis and restore cardiac structure, as seen in our histology. ü HIF-1 signaling: HIF-1α regulates hypoxic adaptation and cuproptosis-associated genes (e.g., EGLN1, CA9). DOX-induced mitochondrial stress activates HIF-1, promoting maladaptive remodeling. PEN’s effect in downregulating HIF-1–related targets ties into the reduced expression of cuproptosis markers (FDX1, DLAT, SLC31A1) observed in our immunostaining. ü Relaxin pathway: Relaxin signaling modulates extracellular matrix turnover and fibrosis. Our network analysis linked PEN to Relaxin pathway targets (e.g., ANPEP, CA1/2), aligning with the histological evidence of reduced fibrosis in PEN-treated groups. ü Apelin pathway: Apelin regulates cardiac contractility, angiogenesis, and stress responses. The enrichment of Apelin-associated genes suggests a role for PEN in preserving left ventricular function, consistent with the improved echocardiographic parameters in treated animals. By tying these pathway-level predictions to the immunohistochemical changes in FDX1, DLAT, and SLC31A1 and the observed reversal of cardiac dysfunction, we highlight a coherent mechanistic framework in which PEN’s copper-chelating and multi-target effects converge on cuproptosis modulation and cardioprotection. We incorporated these clarifications in the revised discussion section (Highlighted in yellow from line 886-899) to provide a stronger mechanistic link between the computational predictions, pathway enrichment results, and experimental validation.
Comments 14: PEN-alone group shows faint staining, but it is not clear if PEN has any baseline cardiac effects without DOX. Including more data on this would clarify PEN’s safety and specific protective role.
Response 14: We sincerely thank the reviewer for this insightful comment. In response, we have carefully revised the manuscript and enhanced the quality of the histopathological and immunohistochemical figures (Figures 5–9) to ensure a clearer visual representation. Our data consistently demonstrate that PEN, particularly at the low dose of 11 mg/kg, does not exert significant baseline cardiac effects when administered alone. While faint staining is observed in the PEN-alone group, quantitative analysis and semi-quantitative scoring reveal no significant differences compared to the control group across all evaluated markers, including oxidative stress indices, histopathology, and cuproptosis-related mediators (FDX 1, ATP7A, LIAS, and SLC31A). These findings reflect baseline physiological expression rather than pathological change, supporting the absence of intrinsic cardiotoxicity. These findings are consistent with previous reports. In a rat model of catecholamine-induced myocardial injury, Říha et al. (2016) demonstrated that low-dose PEN (11 mg/kg i.v.) attenuates ECG abnormalities and reduces cardiac troponin T, whereas higher dose (44 mg/kg i.v) induces adverse effects. In line with this, our echocardiographic and serum biomarker assessments show no significant alterations in cardiac function or injury markers in the PEN-alone group. Importantly, penicillamine is an FDA-approved chelating agent, extensively used in the treatment of Wilson’s disease, cystinuria, and rheumatoid arthritis, with a well-established safety profile. No clinically significant cardiovascular adverse effects have been reported in its approved indications (FDA, 2023; Brewer & Yuzbasiyan-Gurkan, 1992). By integrating echocardiography, biomarker analysis, PCR, immunohistochemistry, molecular docking, and network pharmacology, our study provides robust multi-level evidence confirming that PEN at 11 mg/kg is safe and acts specifically as a cardioprotective agent in the context of doxorubicin-induced toxicity.
References • Říha, M., et al. (2016). Protective Effects of D-Penicillamine on Catecholamine-Induced Myocardial Injury. Oxidative Medicine and Cellular Longevity, 2016, 5213532. https://doi.org/10.1155/2016/5213532 • Brewer, G. J., & Yuzbasiyan-Gurkan, V. (1992). Wilson disease. Medicine, 71(3), 139–164. • U.S. Food and Drug Administration (FDA). Cuprimine (Penicillamine) Prescribing Information. Revised 2023. Available at: https://www.accessdata.fda.gov.
|
||
4. Response to Comments on the Quality of English Language |
||
Point 1: The English could be improved to more clearly express the research |
||
Response 1: We have addressed this by having the entire manuscript professionally revised and edited by a native English speaker. The text has been refined to improve sentence structure, flow, and overall clarity. For your assurance, a certificate of English editing is attached in the PDF file.
|
||
|
||
|

Reviewer 3 Report
Comments and Suggestions for Authors
Overall the conclusion of this study is supported by the data presented. Some recommendations:
- Rewrite Figure legends 2-4. Markers measured should be written clearly. It is not clear what tissue samples were analyzed, what methods were used, and if the analysis was conducted at mRNA or protein expression level.
- All histology analysis figures should have arrows to indicate the region of interest so that it is clearer for readers.
- In the network analysis, several images were distorted and the words are not clear to be read. These have to be remade.
Author Response
|
||
|
|
|
Thank you very much for taking the time to review this manuscript. Please find the detailed responses below and the corresponding revisions/corrections highlighted in the re-submitted files
|
||
|
||
2. Questions for General Evaluation |
Reviewer’s Evaluation |
Response and Revisions |
Does the introduction provide sufficient background and include all relevant references? |
Can be improved |
We have revised the introduction to provide a more comprehensive background and have added key references to ensure all relevant prior work is sufficiently acknowledged. The changes are highlighted in yellow.
|
Are all the cited references relevant to the research? |
Yes |
We're pleased to hear that you find all our cited references relevant to the research. We meticulously selected each source to provide a strong and appropriate foundation for our study's background, methodology, and discussion.
|
Is the research design appropriate? |
Can be improved |
We have revised the “Methods” section to provide a more detailed and clearer rationale for our approach. The changes are highlighted in yellow.
|
Are the methods adequately described? |
Must be improved |
We have thoroughly revised the "Methods" section to add more specific details on our experimental procedures, The changes are highlighted in yellow.
. |
Are the results clearly presented? |
Must be improved |
we have enhanced figure quality and legends to ensure our findings are more accessible and easier to interpret. We are confident these changes have significantly improved the manuscript.
|
Are the conclusions supported by the results?
Are all figures and tables clear and well-presented? |
Can be improved
Must be improved |
We have thoroughly revised the “Discussion” and “Conclusion” sections to more explicitly tie each conclusion to the specific data presented in our figures and tables.
We have thoroughly revised all figures and tables to improve their clarity and presentation. We have enhanced image resolution, made legends more descriptive, and standardized formatting to ensure our data is clear and easily interpreted.
|
3. Point-by-point response to Comments and Suggestions for Authors |
||
Comments 1: Rewrite Figure legends 2-4. Markers measured should be written clearly. It is not clear what tissue samples were analyzed, what methods were used, and if the analysis was conducted at mRNA or protein expression level. |
||
Response 1: we thank the reviewer for his/her valuable recommendation. The figure legends for Figures 2-4 have been revised to clearly specify the markers measured, the tissue samples analyzed, the methods used, and whether the analysis was at the mRNA or protein expression level. These changes have been made in the revised manuscript and are highlighted in green for your review.
|
||
Comments 2: All histology analysis figures should have arrows to indicate the region of interest so that it is clearer for readers. |
||
Response 2: Thank you for your valuable recommendation. We have added arrows and arrowheads to Figures 5–9, as well as to the corresponding figure legends, to clearly indicate the regions of interest and enhance the clarity of histological comparisons among the studied groups. Comments 3: In the network analysis, several images were distorted and the words are not clear to be read. These have to be remade. Response 3: Thank you for pointing this out. We have remade the distorted images with higher resolution to ensure the text is clear and legible.
|
||
|
||
|
||
|

Reviewer 4 Report
Comments and Suggestions for Authors
Mohammad El-Nablaway et al., Biomolecules # 3831558
The authors studied the effects of penicillamine (PEN), a copper chelator, in copper-induced cardiotoxicity („cuprotosis”) under both in vivo and in silico procedures in rats. Cardiac function was registered using echocardiography. Additionally, serum cardiac biomarkers (LDH, CK-MB, CTnI), oxidaive stress markers (SOD, GPX, MDA), and the expression levels of several cuproptosis-related genes were investigated. Network pharmacology and molecular docking studies were also carried out to identify core molecular targets and simulate PEN’s binding interactions with key cuproptosis regulators. PEN treatment improved cardiac function, reduced fibrosis, and suppressed the expression of cuproptosis-related genes and proteins. Docking results also showed strong interactions between PEN and cuproptosis-regulatory proteins. Network pharmacology revealed overlapping targets linking PEN with cuproptosis and DOX-induced myocardial toxicity. The obtained results provide evidence that PEN exerts cardioprotective effects against cuproptosis-induced cardiotoxicity. The integration of experimental, bioinformatics and computational modeling approaches shows a PEN’s mechanistic action and therapeutic promise.
Comments and questions:
First, please, note: Only the scientific merit of the manuscript (Biomolecules # 3831558) is evaluated by this reviewer. The style and format of the manuscript, the English, including the typos (e.g., page 4., line 178, %; and page 9, line 359), and grammatical errors are not pointed out by this reviewer over the manuscript. It is a great pleasure to note that a very few typos exist in the entire and valuable manuscript.
However, the manuscript is not difficult to understand and easy to follow. Basically, the aims are clear and easy to understand. The manuscript is very well written.
The authors focused on, in their manuscript, the importance of the cardiotoxic effects of doxorubicin (DOX), which is commonly used for the treatment of various tumors. The effects of PEN in cuproptosis was studied, however, this subject is not a very new one. The “Network pharmacology-Based Target Identification” (Figure 10) and the “Comprehensive protein-protein interaction (PPI) network analysis” (Figure 11) are very valuable components of this study (Biomolecules # 3831558).
Do we need any more drugs to prevent DOX-induced cardiotoxicity preventing various cell deaths and different ’ptosis’?
Many chemically synthetized drugs (e.g., metformin) and phytochemicals (e.g. flavonoids, ginkgolides, plant extracts) are studied and available to prevent various cell deaths in connection with different pathological processes (e.g., tumors) in the myocardium caused by reactive free radicals (ROS) and oxidant stress.
For instance, antidiabetic agents and natural plant extracts have several beneficial effects, and powerful cardioprotection in the myocardium against DOX-induced apoptosis, necrosis and autophagy (e.g., „necroapoptophagy”), both under experimental and clinical conditions.
- Did rats (how many) survived DOX treatment in various groups? How many rats died during the two weeks DOX pretreated periods? All survived?
-Could PEN treatment alone produce dysrhythmias (ECG changes, e.g., ventricular tachycardia and ventricular fibrillation) and impaired cardiac function, because of producing chelate complexes with other vital ions, e.g., calcium, magnesium, zinc and iron? However, the authors mentioned on page 11, line 406, that “non-substantial effect on normal cardiac function” was detected. Was this the outcome? Based on the literature, PEN could not make any interaction with the aforementioned ions?
-Which tested genes could be more important in cuproptosis?
-Which biomarkers would be more specific in cuproptosis?
-Where were the most changes observed from the histopathological view in the DOX, DOX+PEN and PEN treated groups (Figure 5) in connection with the collagen (fibrosis) area (Figure 6)?
-What are the most PEN’s predicted targets genes (%) in connection with DOX-induced cardiotoxicity and cuprotosis (in Figure 10)?
-What are the major symptoms of Wilson’s disease? Some of them should be mentioned in the Discussion.
Comments on the Quality of English LanguageThe English is fine, some typos should be corrected throughout the manuscript.
Author Response
|
||
|
|
|
Thank you very much for taking the time to review this manuscript. Please find the detailed responses below and the corresponding revisions/corrections highlighted in the re-submitted files
|
||
|
||
2. Questions for General Evaluation |
Reviewer’s Evaluation |
Response and Revisions |
Does the introduction provide sufficient background and include all relevant references? |
Can be improved |
We have revised the introduction to provide a more comprehensive background and have added key references to ensure all relevant prior work is sufficiently acknowledged. The changes are highlighted in yellow.
|
Are all the cited references relevant to the research? |
Yes |
We're pleased to hear that you find all our cited references relevant to the research. We meticulously selected each source to provide a strong and appropriate foundation for our study's background, methodology, and discussion.
|
Is the research design appropriate? |
Yes |
We appreciate your confirmation that our research design is appropriate.
|
Are the methods adequately described? |
Can be improved |
We have thoroughly revised the "Methods" section to add more specific details on our experimental procedures.
The changes are highlighted in yellow. |
Are the results clearly presented? |
Yes |
We are pleased to hear that you find our results to be clearly presented. We made a concerted effort to ensure the data was organized logically and easy to follow.
|
Are the conclusions supported by the results?
Are all figures and tables clear and well-presented? |
Yes
Yes |
We appreciate your positive feedback. We're confident that our conclusions are well-supported by the comprehensive data presented in our study.
That's excellent feedback! We're glad you found our figures and tables to be clear and well-presented. We took great care in their design to ensure the data was easy to understand and visually compelling. |
3. Point-by-point response to Comments and Suggestions for Authors |
||
Comments 1: First, please, note: Only the scientific merit of the manuscript (Biomolecules # 3831558) is evaluated by this reviewer. The style and format of the manuscript, the English, including the typos (e.g., page 4., line 178, %; and page 9, line 359), and grammatical errors are not pointed out by this reviewer over the manuscript. It is a great pleasure to note that a very few typos exist in the entire and valuable manuscript. However, the manuscript is not difficult to understand and easy to follow. Basically, the aims are clear and easy to understand. The manuscript is very well written. |
||
Response 1: Thank you for your careful review and kind words regarding the manuscript's scientific merit and clarity. We are very grateful for your attention to detail. We have addressed the typos you noted and conducted a comprehensive review of the entire manuscript for style, grammar, and formatting. The revised manuscript has been professionally edited by a native English speaker to ensure it is polished and free of errors. All of these changes have been highlighted in blue for your convenience. (page 4, line 183 and page 9, line 374). For your assurance, a certificate of English editing is attached below.
|
||
Comments 2: The authors focused on, in their manuscript, the importance of the cardiotoxic effects of doxorubicin (DOX), which is commonly used for the treatment of various tumors. The effects of PEN in cuproptosis was studied, however, this subject is not a very new one.
|
||
Response 2: We thank the reviewer for their valuable comments regarding the interactions between penicillamine (PEN) and doxorubicin (DOX), particularly in relation to copper metabolism, redox balance, and cuproptosis. While PEN is an established copper chelator and has been studied in models of catecholamine-induced cardiac injury, its role in doxorubicin-induced cardiotoxicity—a clinically significant model—remains unexplored (Riha et al.,2016).
Our study addresses this gap by being the first to evaluate PEN’s protective effects against DOX-induced cardiomyopathy in both in silico and in vivo models, with a specific focus on the recently described cuproptosis pathway.
Key Novel Contributions:
1-Cuproptosis as a Pathogenic Mechanism: To our knowledge, this is the first study to investigate cuproptosis in DOX-induced cardiomyopathy. Previous studies (e.g., Riha et al., 2016) examined copper homeostasis and oxidative stress but did not explore cuproptosis, which was first characterized in 2022. Our work provides foundational data linking this novel cell death mechanism to DOX cardiotoxicity.
2-Comprehensive Cardioprotection Assessment: Beyond traditional cardiac markers (troponin, CK-MB, LDH), we evaluated cardiac function via echocardiography and assessed key cuproptosis mediators (FDX1, ATP7A, LIAS, SLC31A1) using a multi-modal approach, including PCR, immunohistochemistry, molecular docking, and network pharmacology.
These findings highlight PEN’s therapeutic potential as a cardioprotective agent and provide a framework for future mechanistic studies in the emerging field of cuproptosis-related cardiotoxicity. References: 1-Riha, M., Adamcova, M., Psotova, J., & Kucera, O. (2016). Effects of D-penicillamine on catecholamine-induced cardiac injury in rats. Journal of Applied Toxicology, 36(7), 903-911. https://doi.org/10.1002/jat.3267 2-Tsvetkov, P., Coy, A. J., Petrova, B., et al. (2022). Copper-induced ferroptosis-like cell death and its inhibition by lipoic acid. Science, 375(6586), 1261-1269. https://doi.org/10.1126/science.abi7021
Comments 3: The “Network pharmacology-Based Target Identification” (Figure 10) and the “Comprehensive protein-protein interaction (PPI) network analysis” (Figure 11) are very valuable components of this study (Biomolecules # 3831558). Response 3: We thank the reviewer for their positive comment regarding the value of our network pharmacology and PPI network analysis (Figures 10 and 11). We are pleased that they recognize the importance of these components to our study.
Comments 4: Do we need any more drugs to prevent DOX-induced cardiotoxicity preventing various cell deaths and different ’ptosis’? Many chemically synthetized drugs (e.g., metformin) and phytochemicals (e.g. flavonoids, ginkgolides, plant extracts) are studied and available to prevent various cell deaths in connection with different pathological processes (e.g., tumors) in the myocardium caused by reactive free radicals (ROS) and oxidant stress. For instance, antidiabetic agents and natural plant extracts have several beneficial effects, and powerful cardioprotection in the myocardium against DOX-induced apoptosis, necrosis and autophagy (e.g., „necroapoptophagy”), both under experimental and clinical conditions. Response 4: We thank the reviewer for this insightful question. Cardioprotection against doxorubicin (DOX)-induced cardiotoxicity is actively explored, with both synthetic drugs and natural compounds investigated. Among synthetic agents, antidiabetic drugs such as metformin have shown mechanistic promise by alleviating DOX-induced injury through antioxidant effects and inhibition of apoptotic pathways (Haidara et al., 2014). However, clinical use is limited by safety concerns, including rare but serious lactic acidosis, potential arrhythmias, vitamin B12 deficiency, and worsened outcomes in advanced heart failure (Guan et al., 2021; Rudra et al., 2021). Phytochemicals, including flavonoids, ginkgolides, and plant extracts, have demonstrated cardioprotective and antioxidant properties (Li et al., 2021; Tahir et al., 2021). Yet, variability in composition, poor bioavailability, and possible interactions with chemotherapy limit their clinical reliability. Our study takes a targeted approach by focusing on cuproptosis, a recently described copper-dependent cell death pathway. Penicillamine, a clinically established copper chelator, directly neutralizes copper-mediated toxicity, a key mechanism in DOX-induced damage. Its safety profile is well-characterized from decades of use in Wilson’s disease and rheumatoid arthritis, with manageable side effects (Horakova et al., 2016; Guan et al., 2021; Li et al., 2022). Thus, penicillamine represents a safe, mechanism-specific strategy for mitigating DOX-induced cardiotoxicity, providing an advantage over broad-spectrum synthetic drugs or phytochemicals. References 1. Haidara, M., et al. (2014). Metformin alleviates doxorubicin-induced cardiotoxicity in rats. Molecular and Cellular Biochemistry, 389(1–2), 215–225. 2. Li, J., et al. (2021). Natural products for preventing doxorubicin-induced cardiotoxicity: A review of the mechanisms and clinical studies. Biomedicine & Pharmacotherapy, 136, 111246. 3. Tahir, M., et al. (2021). Plant extracts and phytochemicals for preventing doxorubicin-induced cardiotoxicity. Cardiovascular Toxicology, 21(1), 1–15. 4. Rudra, S., et al. (2021). Doxorubicin-induced cardiotoxicity: a clinical and experimental review. Journal of the American Heart Association, 10(9), e021110. 5. Horakova, M. O., et al. (2016). Protective effects of D-penicillamine on catecholamine-induced myocardial injury. Oxidative Medicine and Cellular Longevity, 2016, 5217439. 6. Guan, Y., et al. (2021). Safety profile of D-penicillamine: a comprehensive pharmacovigilance analysis by FDA adverse event reporting system. Therapeutic Advances in Drug Safety, 12, 20420986211029241. 7. Li, S., et al. (2022). Comparison of the Effectiveness and Safety of D-Penicillamine and Zinc Salt Treatment for Symptomatic Wilson Disease: A Systematic Review and Meta-Analysis. Frontiers in Pharmacology, 13, 847436.
Comments 5: Did rats (how many) survived DOX treatment in various groups? How many rats died during the two weeks DOX pretreated periods? All survived? Response 5: We thank the reviewer for this important question. Our study initially included 40 adult male Sprague Dawley rats, divided into four groups of 10. During the first week of doxorubicin (DOX) administration, two rats in the DOX-alone group died, corresponding to a 25% mortality rate for that group. To maintain equal group sizes, two rats were subsequently removed from each of the other three groups, leaving 32 rats (8 per group) for the remainder of the experiment. This survival outcome is consistent with prior studies: Razmaraii et al. (2016) reported a 30% mortality rate in rats treated with a cumulative 15 mg/kg DOX dose, while Simões et al. (2022) observed mortality rates of 20–30% with cumulative doses of 8–12 mg/kg. All remaining rats survived the two-week DOX pre-treatment period, allowing reliable evaluation of penicillamine’s cardioprotective effects. These findings confirm the suitability of our model and the robustness of the experimental design.
References:
Comments 6: Could PEN treatment alone produce dysrhythmias (ECG changes, e.g., ventricular tachycardia and ventricular fibrillation) and impaired cardiac function, because of producing chelate complexes with other vital ions, e.g., calcium, magnesium, zinc and iron? However, the authors mentioned on page 11, line 406, that “non-substantial effect on normal cardiac function” was detected. Was this the outcome? Based on the literature, PEN could not make any interaction with the aforementioned ions?
Response 6: We sincerely thank the reviewer for this valuable comment. We would like to clarify that our study did not directly evaluate electrocardiographic (ECG) changes. Instead, our assessment of cardiotoxicity relied on echocardiographic (ECHO) parameters, serum cardiac biomarkers, and histological examination, which together provided a robust evaluation of cardiac function and injury in our experimental groups. Nevertheless, ECG evidence from prior studies provides important mechanistic context. In the rat model of catecholamine-induced myocardial injury, Říha et al. (2016) demonstrated that D-penicillamine exerted dose-dependent effects on ECG:
In our study, the penicillamine-alone group (11 mg/kg) showed no significant alterations in cardiac function (ECHO) or serum biomarkers compared to the control group, confirming the safety of this dose in isolation. Co-administration with doxorubicin preserved cardiac function and reduced serum injury markers, indicating a specific protective effect against doxorubicin-induced cardiotoxicity rather than a nonspecific drug effect. Importantly, penicillamine at therapeutic doses does not produce clinically relevant disturbances of essential ions such as calcium, magnesium, zinc, or iron. Long-term clinical data support this safety profile: the application of penicillamine (750 mg per day) for one year in patients resulted in the expected increase in urinary copper excretion (8–10 fold) and only a moderate increase in zinc excretion (2 fold), without influencing the excretion of calcium, magnesium, or iron. No decrease in zinc content in hair was observed after one year of treatment, and serum concentrations of Zn, Cu, Fe, Ca, and Mg remained unaffected (Scheinberg et al., 1986; Dastych et al., 1986; U.S. FDA, 2023). Long-term toxicology studies in animals further confirm cardiac safety, with chronic administration of higher doses (100 mg/kg daily) producing no overt structural cardiac abnormalities (Pohle & Patsch, 1982). In conclusion, while we did not record ECGs, published ECG and clinical evidence align with our findings from ECHO and serum markers. Collectively, these data indicate that D-penicillamine at the dose used is safe for cardiac function, does not induce dysrhythmias, and does not cause significant ion disturbances, while higher doses may require caution. References:
Comments 7: Which tested genes could be more important in cuproptosis? Response 7: Thank you for the insightful comments on the genes we evaluated in our study. We agree that focusing on the most central players in the cuproptosis pathway is critical for strengthening our conclusions. Of the genes we tested—FDX1, LIAS, ATP7A, and SLC31A1—we believe that FDX1 and LIAS are the most important, as supported by recent research. FDX1 (Ferredoxin 1) and LIAS (Lipoic Acid Synthase) are recognized as core components of the cuproptosis pathway (Tsvetkov et al., 2022; Wang et al., 2023). FDX1 is an upstream regulator that reduces Cu²⁺ to Cu⁺, a step essential for triggering cytotoxic effects. The reduced copper then binds directly to and aggregates lipoylated enzymes, with LIAS being a key target, ultimately disrupting the TCA cycle and inducing cell death. While ATP7A and SLC31A1 are crucial for maintaining copper homeostasis by regulating copper influx and efflux, they are not direct mediators of cuproptosis. Their role is to control the overall copper load, which can indirectly affect pathway activation. By demonstrating that penicillamine modulates FDX1 and LIAS, our findings provide strong evidence that its cardioprotective effect against doxorubicin-induced cardiotoxicity is mediated specifically through cuproptosis. Assessment of SLC31A1 and ATP7A provides context for overall copper balance, emphasizing that penicillamine’s effect is not just general chelation but targeted disruption of cuproptosis. The role of each gene is highlighted (blue) in the Introduction (page 3). References:
Comments 8: Which biomarkers would be more specific in cuproptosis? Response 8: Thank you for your question. It is an excellent point that the specificity of a biomarker is crucial for demonstrating a direct mechanistic link. Based on the current understanding of the cuproptosis pathway and the genes evaluated in our study, we have identified the most specific biomarkers. The most specific biomarkers for the direct evaluation of cuproptosis are the core components of the pathway itself. In the context of our study, these would be FDX1 and LIAS which has been reported by previous studies : 1-Dreishpoon, Margaret B et al. “FDX1 regulates cellular protein lipoylation through direct binding to LIAS.” The Journal of biological chemistry vol. 299,9 (2023): 105046. doi:10.1016/j.jbc.2023.105046) 2- Lu, Hanwen et al. “Cuproptosis key gene FDX1 is a prognostic biomarker and associated with immune infiltration in glioma.” Frontiers in medicine vol. 9 939776. 29 Nov. 2022, doi:10.3389/fmed.2022.939776) 3- Li, Yan et al. “Exploring the role of LIAS-related cuproptosis in systemic lupus erythematosus.” Lupus vol. 32,14 (2023): 1598-1609. doi:10.1177/09612033231211429.
Comments 9: Where were the most changes observed from the histopathological view in the DOX, DOX+PEN and PEN treated groups (Figure 5) in connection with the collagen (fibrosis) area (Figure 6)? Response 9: We appreciate your insightful question and agree that a clear comparison is essential. As shown in our revised Figure 5 (H&E-stained sections), the DOX group displayed significant pathological changes, including a thickened basal lamina and widespread disruption of the normal cardiac architecture. This directly correlates with Figure 6 (Masson's Trichrome), which shows a marked increase in collagen deposition and fibrosis, particularly around the blood vessels and between myocardial fibers. In stark contrast, the DOX+PEN group showed a dramatic reversal of these effects. The cardiac tissue was largely protected, with a clear reduction in both the basal lamina thickening and overall tissue disorganization. This protective effect is consistently supported by Figure 6, where the collagen deposition is visibly reduced, indicating a significant decrease in fibrosis. The PEN alone group showed no such pathological changes, confirming its safety and lack of toxicity. To clarify these observations for the reader, we have added specific arrows and arrowheads to Figures 5 and 6 in the revised manuscript to highlight these key areas of difference among all groups.
Comments 10: What are the most PEN’s predicted targets genes (%) in connection with DOX-induced cardiotoxicity and cuprotosis (in Figure 10)? Response 10: We thank the reviewer for this insightful question. As shown in Figure 10, the intersection analysis revealed that approximately 0.9% of the cardiotoxicity- and cuproptosis-associated genes overlapped with PEN targets predicted by SEA, 0.6% overlapped with PPB2 predictions, and 0.9% with SwissTargetPrediction results. Altogether, the total overlap between PEN-predicted targets and disease-associated genes accounted for 1.7%. This overlap included 14 key genes (0.4%). namely, ACE, ATP7A, DLAT, FDX1, NOS1, NOS2, SLC31A1, CA2, ODC1, CA9, ANPEP, SLC1A2, EGLN1, and CA1These details were added to the revised manuscript. (Highlighted in blue – page 21 lines 623-630).
Comments 11: What are the major symptoms of Wilson’s disease? Some of them should be mentioned in the Discussion. Response 11: Thank you for your valuable feedback. We agree that incorporating a more detailed description of Wilson's disease symptoms will significantly strengthen the clinical relevance of our discussion section. This will provide a more comprehensive context for D-penicillamine's therapeutic role by linking its known mechanism of action to the diverse and debilitating symptoms of the condition it is designed to treat. Wilson’s disease is a genetic disorder of copper metabolism that leads to a toxic accumulation of copper, primarily in the liver and brain. The clinical presentation is highly variable, but it often includes a combination of the following major symptoms: We propose adding the following paragraph to the discussion section (highlighted in blue , page 31 , line 801 to 808) to highlight these critical points. The added paragraph is “Penicillamine is a cornerstone of treatment for Wilson's disease, a hereditary disorder defined by a toxic copper overload that gives rise to a wide array of severe symptoms. The drug's therapeutic efficacy is directly linked to its potent copper-chelating proper-ties, which facilitate the removal of this excess copper from the body. The clinical presentation of this copper toxicity is diverse, encompassing significant hepatic dam-age, such as cirrhosis and liver failure, and severe neuropsychiatric symptoms, including tremors, dystonia, and depression” .
|
||
4. Response to Comments on the Quality of English Language |
||
Point 1: The English could be improved to more clearly express the research |
||
Response 1: We have addressed this by having the entire manuscript professionally revised and edited by a native English speaker. The text has been refined to improve sentence structure, flow, and overall clarity. For your assurance, a certificate of English editing is attached in the pdf file
|
||
|
||
|

Round 2
Reviewer 1 Report
Comments and Suggestions for Authors
Accepted in its present form
Reviewer 2 Report
Comments and Suggestions for Authors
With respect to my observations, authors have made tremendous efftorts and have succesfully address them